# A 3D-Var Assimilation Scheme for Vertical Velocity with the CMA-MESO v5.0

Hong Li[1,2], Yi Yang[1], Jian Sun[3], Yuan Jiang[3], Ruhui Gan[1], Qian Xie[1]

[1]Key Laboratory of Climate Resource Development and Disaster Prevention in Gansu Province, Center for Weather Forecasting and Climate Prediction of Lanzhou University, College of Atmospheric Sciences, Lanzhou University, Lanzhou 730000, China
[2]Institute of Arid Meteorology, China Meteorological Administration, Lanzhou 730020, China
[3]China Meteorological Administration Earth System Modeling and Prediction Centre, Beijing 100081, China

*Correspondence to*: Yi Yang (yangyi@lzu.edu.cn)

**Abstract.** Certain vertical motions associated with meso-microscale systems are favorable for convection development and maintenance, correct initialization of updraft motions is thus significant in convective precipitation forecasts. A three-dimensional variational-based vertical velocity ($w$) assimilation scheme has been developed within the high-resolution (3 km) CMA-MESO (the Mesoscale Weather Numerical Forecast System of China Meteorological Administration) model. This scheme utilizes the adiabatic Richardson equation as the observation operator for $w$, enabling the update of horizontal winds and mass fields of the model's background. The tangent linear and adjoint operators are subsequently developed and undergo an accuracy check. A single-point $w$ observation assimilation experiment reveals that the observational information is effectively spread both horizontally and vertically. Specifically, the assimilation of $w$ contributes to the generation of horizontal wind convergence at lower model levels and divergence at higher model levels, thereby adjusting the locations of convection occurrence. The impact of assimilating $w$ on the forecast is then examined through a series of continuous 10-day runs. Further assimilation of $w$, in addition to the assimilation of conventional and radial wind data, significantly improves the forecast accuracy of precipitation, resulting in higher FSS (frequency skill score) values and higher ETS (equitable threat score) skills at higher thresholds (5 and 20 mm h$^{-1}$). However, it should be noted that further assimilation of $w$ can potentially lead to some false precipitation, resulting in slightly lower ETS values at lower thresholds (1 mm h$^{-1}$) and a neutral impact on BIAS (bias score) skills. An individual case study conducted within the batch experiments reveals that assimilating $w$ has a beneficial impact on the enhancement of vertical motion across different layers of the model, facilitating the transport of moisture from lower to mid-high model levels, thereby leading to an improvement in forecast skills.

## 1 Introduction

The vertical component of atmospheric motion plays a pivotal role in the definition of convection, as it directly influences the formation and development of clouds along with their associated precipitation. In numerical models, vertical motions are of utmost importance in parameterizing cloud dynamics and microphysical processes. This significance stems from their

ability to describe the coupling between atmospheric dynamics and cloud formation and development. Consequently, they hold a crucial position in forecasting convective-scale precipitation (see, e.g., Donner et al., 2001; Lang et al., 2007; Panosetti et al., 2019; Tao et al., 2022). A three-dimensional analysis field that accurately involves both updrafts and downdrafts holds significant promise for improving the accuracy of convective precipitation forecast.

The vertical velocity ($w$) is difficult to measure directly or estimate due to its transient nature and relatively small magnitude, which is typically a few orders of magnitude smaller than the mesoscale horizontal velocities (Lee et al., 2003; Tarry et al., 2022). The well-known direct measurement is the research aircraft (e.g., LeMone and Zipser, 1980; Houze and Betts, 1981; Rodts et al., 2003; Anderson et al., 2005; May et al., 2008; Heymsfield et al., 2010) but with limited spatial and temporal coverage. Besides, the nature of vertical velocities allows them can be inferred from balanced dynamics. The widely

acknowledged inference of such is the so-called "continue equation", from which the $w$ pseudo-observations are derived from horizontal divergence or convergence (Bellamy, 1949; Cifelli et al., 1996). Based on the above principle or other algorithms (e.g., Williams, 2012), $w$ values can also be retrieved from remote sensing instruments, such as wind profilers and scanning Doppler radars (e.g., Lee et al., 2003; Liu et al., 2005; Lee et al., 2006; Heymsfield et al., 2010; Giangrande et al., 2013; Ovchinnikov et al., 2019). Motivated by the development of observation instruments and inversion algorithms, an

increasing number of updraft and downdraft velocities emerged, especially at the cloud-resolve scale (e.g., Doppler radar and lightning data), so it is necessary to evaluate the effects of $w$ assimilation on convective-scale precipitation forecasting.

In fact, efforts have been made to assimilate dynamic information associated with atmospheric vertical motions in recent research. For example, the $w$ information retrieved from lightning data was assimilated based on the well-defined correlation (Price and Rind, 1992) between the total lightning flash rate and the updraft velocities (Wang et al., 2020; Xiao et al., 2021;

Gan et al., 2021). These studies have shown that the assimilation of $w$ improves the water vapor and dynamic fields and therefore produces better forecasts of convective precipitation. It should be noted that another work by Gan et al. (2022) revealed that the assimilation of the "zero" column maximum $w$ can also effectively suppress spurious convections by weakening vertical motions and reducing the hydrometeors and humidity of the model. The above $w$ assimilation attempts are based on nudging (Wang et al., 2020), four-dimensional variational (Xiao et al., 2021), or ensemble square root filter

(Gan et al., 2021, 2022) methods, which are 1) relatively difficult to apply into the operational mesoscale regional models for computational cost consideration or 2) lack of strict physical constraints.

Since the three-dimensional variational (3D-Var) method is still widely used in operations (Gustafsson et al., 2018) due to its lower computational costs and the ability to assimilate nonmodal variables, the development of a 3D-Var assimilation technique for $w$ observation becomes necessary. Within the 3D-Var framework, assimilating $w$ faces numerous difficulties,

the most challenging of which is the establishment of an effective assimilation method that produces a reasonable positive impact on forecasts. By extending $w$ as a control variable, direct assimilation of $w$ becomes feasible, simplifying the observation operator into a mapping algorithm from the model space to the observation space. However, as noted by Chen et al. (2020), the imbalance between microphysical and dynamic fields may lead to excessive noise when directly assimilating $w$ observations with the control variable $w$, accomplished by adding an observation term to the 3D-Var cost function. To

address this issue, Chen et al. (2020) initially computed horizontal convergence (based on the mass-continuity equation) from $w$ pseudo-observations derived from total lightning data. Subsequently, an observation operator for horizontal convergence was developed. To achieve direct assimilation of $w$ while mitigating noise, a transformation observation operator, often referred to as the observation operator, is required. This operator ensures adherence to physical constraints and links the $w$ observations to other state variables of the model to minimize the 3D-Var cost function. In this study, the

adiabatic Richardson equation (Richardson, 1922) is used as the observation operator of $w$. This choice enables the simultaneous update of dynamical and mass fields, thereby promoting a more balanced final analysis. Additionally, this direct assimilation scheme avoids the inversion errors associated with an indirect assimilation approach.

In this study, a 3D-Var assimilation scheme for $w$ is established within the Mesoscale Weather Numerical Forecast System of China Meteorological Administration (CMA-MESO) model. The following is the outline of this study: A brief description

of the basic formulation of 3D-Var and the assimilation strategy for $w$ observation is presented in Sect. 2. In Sect. 3, a single-point observation experiment is performed to test the spread of observational information of the assimilation scheme. The effect of assimilating $w$ observations is then assessed by a series of continuous 10-day runs and an individual case within it, and the results are presented and discussed in Sect. 4. Finally, the main conclusions are addressed in Sect. 5.

## 2 Assimilation system and vertical velocity assimilation strategy

### 2.1 CMA-MESO 3D-Var system

In this study, the CMA-MESO model version 5.0 is used as the forecast model. CMA-MESO (Shen et al., 2020) is a non-hydrostatic regional mesoscale system with a horizontal resolution of 3 km. The $w$ assimilation scheme is constructed within the 3D-Var framework of the CMA-MESO model. In the traditional framework of a variational assimilation system, the best analysis $\mathbf{x}$ can be derived from the control variable $\mathbf{c_v}$ (the control variables for CMA-MESO include the zonal and

meridional winds, pseudo-relative humidity, temperature, and surface pressure) by minimizing a cost function $J$ of $\mathbf{c_v}$ (Courtier et al., 1994):

$$J(\mathbf{c_v}) = \frac{1}{2}\mathbf{c_v}^\mathrm{T}\mathbf{c_v} + \frac{1}{2}(\mathbf{HUc_v} + \mathbf{d})^\mathrm{T}\mathbf{R}^{-1}(\mathbf{HUc_v} + \mathbf{d}), \tag{1}$$

$$\mathbf{x}=\mathbf{x^b}+\mathbf{Uc_v}, \tag{2}$$

where $\mathbf{x^b}$ is the background field, $\mathbf{R}$ is the observation error covariance, $\mathbf{d}$ is defined as $H(\mathbf{x^b})$-$y^o$ ($H$ denotes the observation

operator and $y^o$ is the observation), $\mathbf{H}$ is the linearized observation operator, and $\mathbf{U}$ is associated with the background error covariance $\mathbf{B}$: $\mathbf{B}=\mathbf{U}^\mathrm{T}\mathbf{U}$. The matrix $\mathbf{B}$ is statistically based on the National Meteorological Center method (Parrish and Derber, 1992).

## 2.2 Observation operator for vertical velocity

The observation operator $H$ is used to derive the model equivalent of the observations from the model state variables (Kalnay, 2002). In this study, the adiabatic Richardson equation (Richardson, 1922) is used as the observation operator:

$$\gamma P \frac{\partial w}{\partial z} = -\gamma P \nabla \cdot V_h - V_h \cdot \nabla P + g \int_z^\infty \nabla \cdot (\rho V_h) dz, \tag{3}$$

where $\gamma$ is the ratio of specific heat capacities of air at a constant pressure ($c_p$) and at a constant volume ($c_v$), $P$ is pressure, $z$ is the height, $V_h$ is the horizontal wind (components $u$ and $v$), $g$ is the acceleration due to gravity and $\rho$ is density. The Richardson equation combines the continuity equation, adiabatic equations, and hydrostatic relation, which enables the 3D-Var method to adjust the dynamic and mass fields simultaneously and results in a more balanced analysis field. As the terrain-following vertical coordinate (Gal-Chen and Somerville, 1975) used in the CMA-MESO model is expressed as:

$$\hat{z} = z_T \frac{z - Z_s(x,y)}{Z_T - Z_s(x,y)}, \tag{4}$$

here, $z_T$ is the top height of the model upper boundary and $z_s$ is the topographic height, the Eq. (3) in the terrain-following vertical coordinate can be expressed as:

$$\gamma \Pi^\kappa \frac{\Delta Z_s}{Z_T} \frac{\partial \hat{w}}{\partial z} = -\left(u \frac{\partial \Pi^\kappa}{\partial x} + v \frac{\partial \Pi^\kappa}{\partial y}\right) + \frac{\Delta Z_z}{\Delta Z_s}\left(\frac{\partial Z_s}{\partial x} + \frac{\partial Z_s}{\partial y}\right)\frac{\partial \Pi^\kappa}{\partial z} - \gamma \Pi^\kappa \left(\frac{\partial u}{\partial x} + \frac{\partial v}{\partial y}\right) - \gamma \Pi^\kappa \frac{\partial}{\partial z}\left(\frac{\Delta Z_z}{\Delta Z_s}\right)\left(u \frac{\partial Z_s}{\partial x} + v \frac{\partial Z_s}{\partial y}\right) - \int_z^\infty \frac{\partial \Pi^\kappa}{\partial z}\left(\frac{\partial u}{\partial x} + \frac{\partial v}{\partial y}\right) dz -$$

$$\int_z^\infty \left(u \frac{\partial}{\partial x}\left(\frac{\partial \Pi^\kappa}{\partial z}\right) + v \frac{\partial}{\partial y}\left(\frac{\partial \Pi^\kappa}{\partial z}\right)\right) dz + \int_z^\infty \frac{\Delta Z_z}{\Delta Z_s} \frac{\partial \Pi^\kappa}{\partial z}\left(\frac{\partial Z_s}{\partial x}\frac{\partial u}{\partial z} + \frac{\partial Z_s}{\partial y}\frac{\partial v}{\partial z}\right) dz + \int_z^\infty \frac{\Delta Z_z}{\Delta Z_s} \frac{\partial}{\partial z}\left(\frac{\partial \Pi^\kappa}{\partial z}\right)\left(u \frac{\partial Z_s}{\partial x} + v \frac{\partial Z_s}{\partial y}\right) dz, \tag{5}$$

where $u$ and $v$ are the zonal and meridian wind components, respectively. $\Pi$ is the dimensionless pressure, and $\Pi = \left(\frac{P}{P_0}\right)^{R/c_p}$, $P_0 = 1000$ hPa, and $R$ is the gas constant. The parameter $\kappa$ in Eq. (5) can be expressed as $\kappa = {c_p}/{R}$. $\Delta Z_s$ and $\Delta Z_z$ in Eq. (5) are defined as follows:

$$\Delta Z_s = Z_T - Z_s(x,y), \tag{6}$$

$$\Delta Z_z = Z_T - z(x,y), \tag{7}$$

$\hat{w}$ in Eq. (5) is the $w$ under the vertical coordinate that follows the terrain and is expressed as:

$$\hat{w} = \frac{d\hat{z}}{dt} = \frac{Z_T}{\Delta Z_s}\left(w - \frac{\Delta Z_z}{\Delta Z_s}w_s\right), \tag{8}$$

where $w_s$ is the $w$ value at the surface and $w_s = u\frac{\partial Z_s}{\partial x} + v\frac{\partial Z_s}{\partial y}$.

The observation operator links the $w$ variable with the $u$, $v$, and $\Pi$ variables. $u$ and $v$ are control variables, and $\Pi$ is related to the surface pressure (control variable). Thus, as $w$ is assimilated through Eq. (5), the $w$ of the initial field is not updated directly, but the horizontal winds and pressure fields are updated. Since the $w$ observation term is added as a new kind of observation to the cost function of the 3D-Var system within the CMA-MESO model, modifications made to the existing 3D-Var system include the following: 1) the observation operator for $w$ is established to calculate observation innovation; 2) the tangent linear of the observation operator and its adjoint for the $w$ term are included to calculate the cost function and its gradient values.

## 2.3 Accuracy check

After completion of the $w$ observation operator, the correctness of the adjoint operator should be checked (adjoint check). For the tangent linear $\mathbf{H}$ and its adjoint $\mathbf{H}^{\mathrm{T}}$ of an observation operator, the following formula is always satisfied:

$$< \mathbf{H}(\delta x), \mathbf{H}(\delta x) >=< \mathbf{H}^{\mathrm{T}}(\mathbf{H}(\delta x)), \delta x >, \tag{9}$$

where $\delta x$ represents a small perturbation and $<>$ stands for the inner product of the vectors. The difference between the left-hand side and the right-hand side of the Eq. (9) is expected to approach zero, typically with at least 13 significant digits. The test results show that the term $< \mathbf{H}(\delta x), \mathbf{H}(\delta x) >$ is equal to 0.100159014620902D-17 (D: double precision), and the term $< \mathbf{H}^{\mathrm{T}}(\mathbf{H}(\delta x)), \delta x >$ is equal to 0.100159014620902D-17. The difference between the two terms is 0.577778983316171D-33, which is achieved with 16 digits of accuracy. As a result, the adjoint check has successfully passed under double precision.

For a tangent linear operator, it is also necessary to verify the correctness of the gradient (gradient check) using the following standard:

$$\Phi(\alpha) = \frac{J(\mathbf{c_v}+\alpha)-J(\mathbf{c_v})}{\alpha \nabla J(\mathbf{c_v})}, \tag{10}$$

$$\lim_{\alpha \to 0} \Phi(\alpha)=1, \tag{11}$$

where $\nabla J$ is the gradient of $J$ and the symbol $\alpha$ indicates a small value. For values of $\alpha$ that are near but not too close to the machine zero, the value of $\Phi(\alpha)$ is expected to be close to 1. The results of the gradient check are presented in Table 1, showing a satisfactory approximation of the gradient with 8 digits of accuracy achieved ($\alpha=10^{-7}$). This suggests that the tangent linear operator is accurate within the rounding error of the computer.

Table 1. Verification of gradient correctness: values of $\Phi(\alpha)$ for different $\alpha$ values (symbols defined in Eq. (10)).

| $\alpha$ | $\Phi(\alpha)$ |
| --- | --- |
| $10^{-4}$ | 1.00000684582308 |
| $10^{-5}$ | 1.00000068454433 |
| $10^{-6}$ | 1.00000006939151 |
| $10^{-7}$ | 1.00000000569911 |
| $10^{-8}$ | 1.00000003421803 |
| $10^{-9}$ | 1.00000055706492 |
| $10^{-10}$ | 1.00000626084813 |
| $10^{-11}$ | 1.00001576715311 |
| $10^{-12}$ | 1.00053861393468 |
| $10^{-13}$ | 0.998162037654562 |

## 3 Single-point observation experiment

To investigate the spatial propagation of pseudo-observation information for variable $w$, a single $w$ pseudo-observation is assimilated to assess the changes in various variables. This pseudo-observation of $w$ is positioned at an altitude of 5448.6 m (23rd model level, approximately 500 hPa) at coordinates (38.0° N, 115.2° E) (depicted as solid white or black dots in Fig. 1) with a value of 1 m s$^{-1}$. The observation error is set to 0.5 m s$^{-1}$. The $w$ value of the background field at this location is -0.04 m s$^{-1}$, resulting in an innovation (observation minus background) of approximately 1.04 m s$^{-1}$.

The analysis increment induced by this observation is depicted in Fig. 1. The computed analysis increments of the horizontal wind field and its convergence at the 13th (~850 hPa) or 27th (~400 hPa) model level exhibit an isotropic structure centered around the observation site. Since a positive $w$ value is assimilated, a horizontal wind convergence increment is observed at the lower (13th) model level, while a horizontal wind divergence increment occurs at the middle (27th) model level. The increments of horizontal wind at the lower and middle model levels can reach 0.060 and 0.077 m s$^{-1}$, respectively. As the $w$ observation operator is not directly related to temperature and specific humidity but is rather related to dimensionless air pressure, adjustment of temperature and humidity is achieved through weak physical constraints, resulting in relatively small increments in temperature (~ $-8.7 \times 10^{-6}$–$2.6 \times 10^{-5}$ K in Fig. 1 (d)) and specific humidity (~ $-7.0 \times 10^{-8}$–$6.4 \times 10^{-8}$ kg kg$^{-1}$ in Fig. 1 (c)). From the vertical cross-section of the analysis increment for each variable (Fig. 2), it can be seen that the increase in specific humidity is primarily concentrated in the lower layer below the observation location, while the increases in the other three variables are distributed throughout the entire layer. Regarding the increase in horizontal wind component $u$, below the single point observation, there is a convergence of $u$ wind that extends to 1000 hPa. Above the single point observation, there is a divergence of $u$ wind that extends to approximately 150 hPa. It should be noted that there are currently no constraints on the propagation of the impact of the $w$ assimilation in the vertical direction. However, it is better to set limits to prevent excessive increments at higher model levels, which leads to more realistic forecasts.

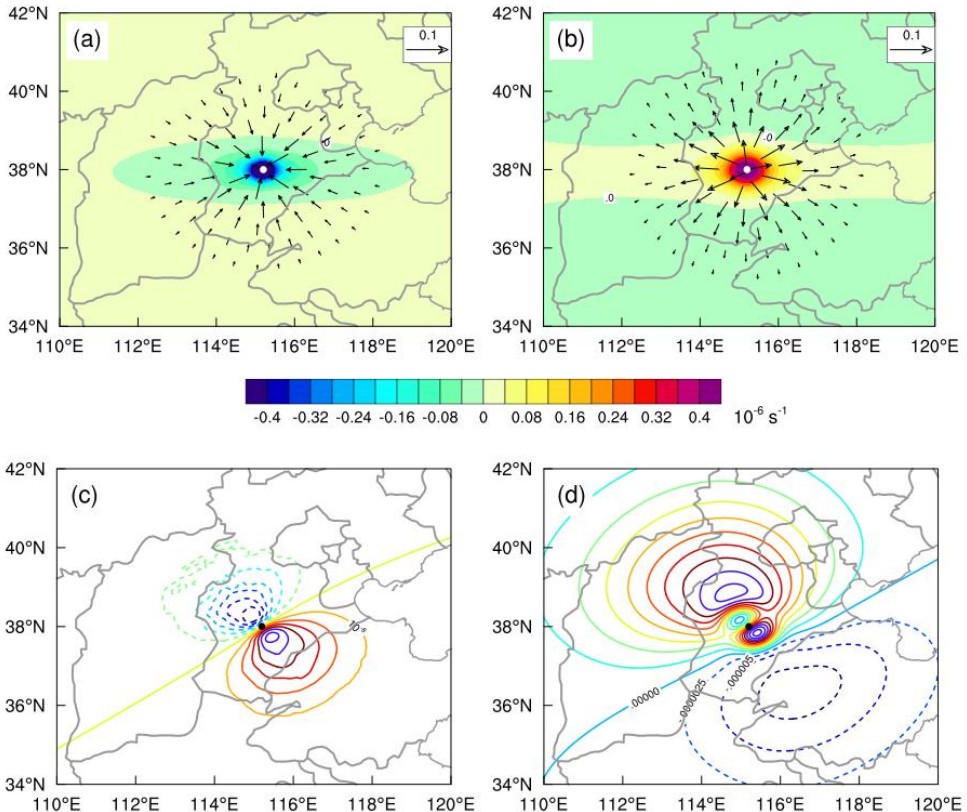

**Figure 1: Analysis increments of different variables at 1500 UTC on July 4, 2020, for the single observation experiment. (a) The horizontal wind (vector; only values greater than 0.01 are shown; unit: m s⁻¹) and horizontal wind divergence (color; unit: 10⁻⁶ s⁻¹) increments at the 13th (~850 hPa) level of the model; (b) is the same as (a) but for the 27th (~400 hPa) level of the model. (c) Specific humidity (interval is 10⁻⁸; unit: kg kg⁻¹) and (d) temperature (interval is 2.5×10⁻⁶ K) increments at the 13th level of the model. The solid white (black) dots in (a) and (b) ((c) and (d)) indicate the locations of the single _w_ observation (38.0° N, 115.2° E).**

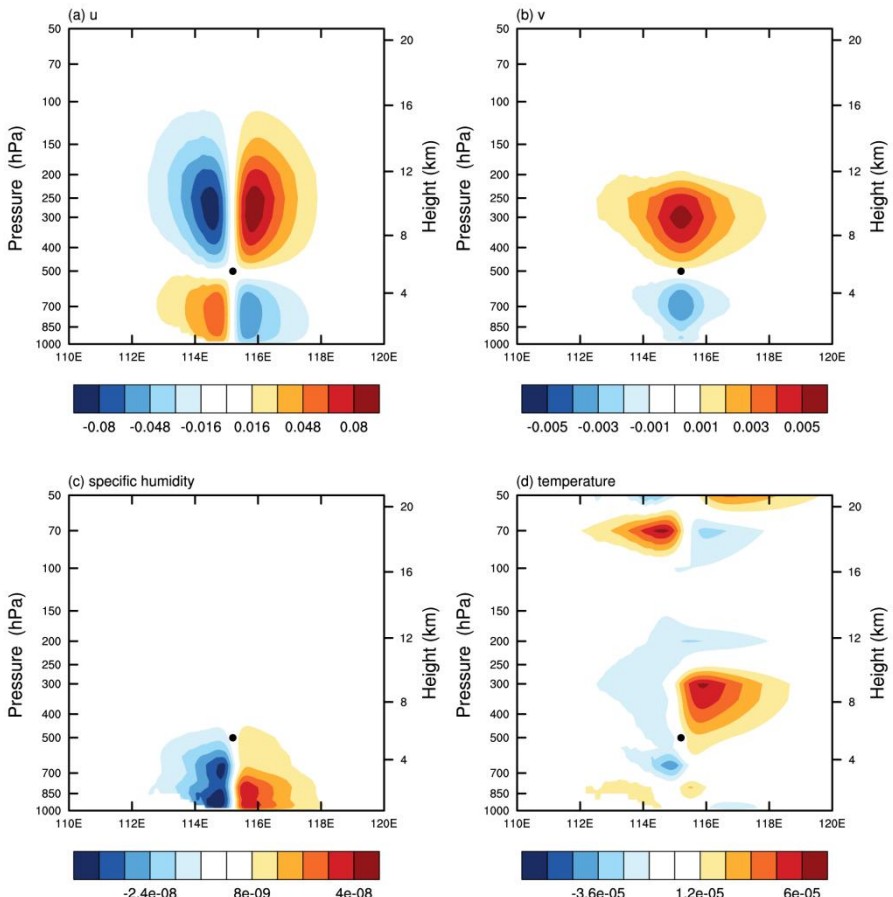

Figure 2: Analysis increments of (a) zonal wind $u$, (b) meridional wind $v$ (unit: m s$^{-1}$), (c) specific humidity (unit: kg kg$^{-1}$), and (d) temperature (unit: K) in a vertical cross-section at 38.0° N at 1500 UTC on July 4, 2020, for the single observation experiment. The solid black dots in the figure represent the locations of the single observation.

## 4 Validation

In this section, a series of runs are conducted from July 1 to 10, 2020, to evaluate the influence of assimilating $w$ observations on convective precipitation forecasting. Especially, the case that took place on July 9, 2020, from the batch experiments is utilized to have a further study.

## 4.1 The pseudo-$w$ observations and precipitation observations

The pseudo-$w$ observations used in this section are derived from radar reflectivity data. Notably, the scope of the assimilation experiment can be extended to encompass $w$ what is observed or retrieved from alternative sources. The radar data used to derive pseudo-observations of $w$ are sourced from the China Next-Generation Weather Radar (CINRAD) network and subjected to quality control procedures. Radar reflectivity serves as an indicator of convection

intensity, while $w$ determines the vigor of convection. Radar reflectivity encompasses information about updraft motion, making it suitable for deriving pseudo-observations of $w$. Given that the vertical profile of $w$ within the convective zone assumes a parabolic shape (Yuter and Houze, 1995; Collois et al., 2013; Schumacher et al., 2015), empirical Eq. (12), as utilized by Liu et al. (2010), can be employed to derive pseudo-$w$ observations.

$$w = (\alpha \times (Z - Z_0) + \beta) \times e^{-(\lambda \times (H_{ei} - H_0))^2}, \tag{12}$$

here, $\alpha$, $\beta$, and $\lambda$ represent coefficients, with $\alpha$ and $\beta$ set to 0.1 and 0.3 respectively, in accordance with Liu et al. (2010). The coefficient $Z_0$ (which is 35 dBZ in this study) denotes the minimum reflectance factor value employed for $w$ retrieval, and $Z$ represents the reflectivity factor, which is larger than $Z_0$. $H_{ei}$ is the height of the observation and $H_0$ signifies the height (unit: km) at which the maximum value of $w$ is attained, while $\lambda$ defines the primary distribution range of $w$ in the vertical direction.

The precipitation observations used to evaluate the model forecast performance are sourced from a merged hourly 0.1°×0.1° precipitation grid dataset, combining data from China's automatic stations and the Climate Prediction Center morphing technique (CMORPH) satellite precipitation data.

## 4.2 The batch experiment

To assess the assimilation impact of $w$ pseudo-observations, a series of continuous 10-day runs spanning from July 1 to July 10, 2020, were conducted. The simulation area corresponds to the operational area of the CMA-MESO model (refer to Fig. 3 (a)), centered at coordinates (35.05° N, 107.5° E). The horizontal grid comprises 2501×1671 grid points with a grid spacing of 0.03° (~3 km). The vertical dimension is represented by 49 levels extending to a model top of 35 km. Initial and lateral boundary conditions are from the National Centers for Environmental Prediction (NCEP) Global Forecast System (GFS) data. The WSM 6-class microphysics scheme (Hong and Lim, 2006), Dudhia shortwave radiation scheme (Dudhia, 1989), Rapid Radiative Transfer Model (RRTM) longwave radiation scheme (Mlawer et al., 1997), and New Medium Range Forecast (NMRF) planetary boundary layer scheme (Han and Pan, 2006) are adopted. Additionally, the cumulus parameterization is closed in these simulations.

### 4.2.1 Experimental design

Two sets of distinct experiments were configured, and an overview of the experimental setup is depicted in Fig. 3 (b). Both the CTRL and DA-W experiments were initialized at 0000 UTC daily from July 1 to July 10, 2020, and ran until 1200 UTC each day. The first 3 hours were considered as the "spin-up" period. In the CTRL experiments, observations from aircraft measurements, radiosondes, and other sources (for a comprehensive list, refer to Fig. 3 (b)) were assimilated from 0300 to 0600 UTC with a 1-hour assimilation interval (radial velocity observations are available at each analysis time, while other data sources are only available at 0300 and 0600 UTC). The CTRL-1CY experiment indicates assimilation at 0300 UTC only, while the CTRL-2CY experiment represents assimilation at 0300 and 0400 UTC, and so on (the number preceding the

experiment name "CY" represents the assimilation iterations). The DA-W experiments are similar to the CTRL experiments but include the assimilation of *w* pseudo-observations (*w* pseudo-observations are available at each analysis time).

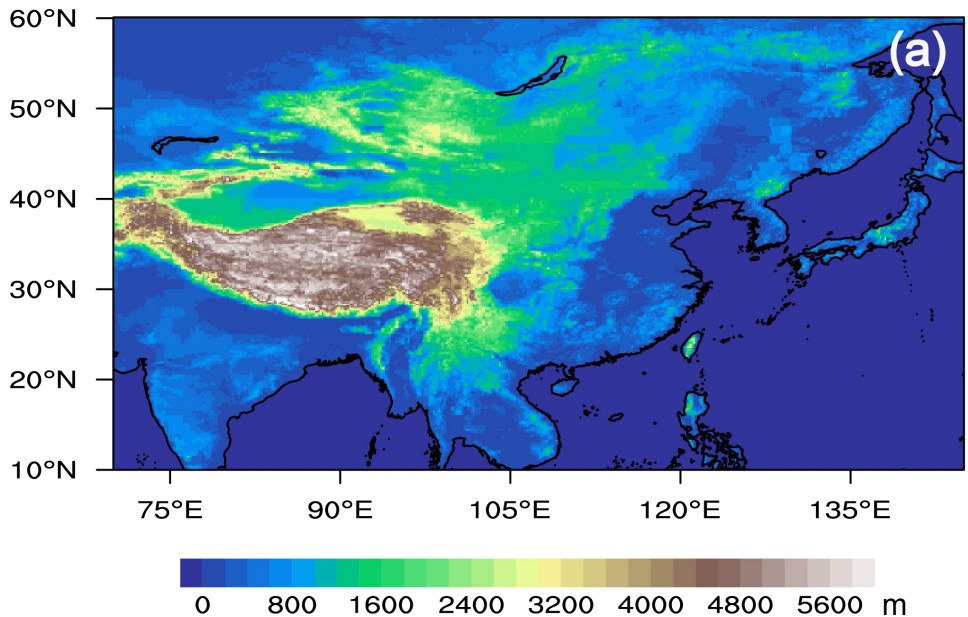

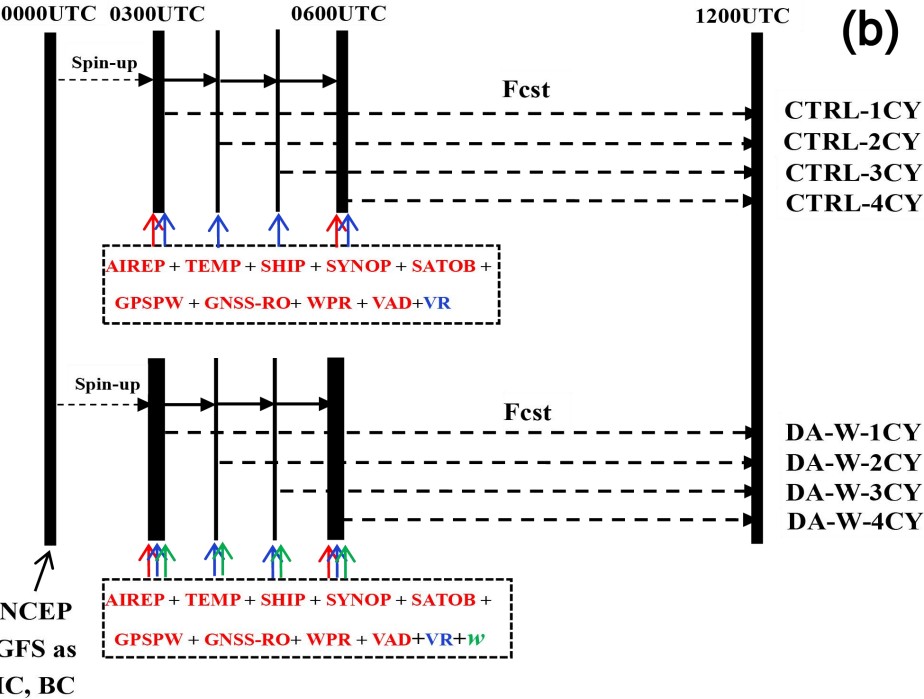

Figure 3: Simulation domain (a) and numerical experimental scheme (b) for the CTRL and DA-W batch experiments. Both experiments utilize NCEP GFS data as the initial condition (IC) and boundary condition (BC). The abbreviation "Fcst" represents forecast. The assimilated data comprises conventional observations from aircraft measurements (AIREP), radiosondes (TEMP),

ships (SHIP), and ground stations (SYNOP). In addition, cloud-track-wind (SATOB), precipitable water derived from the Global Positioning System (GPSPW), refractivity radio-occultation data from the Global Navigation Satellite System (GNSS-RO), wind profiler radar (WPR), velocity-azimuth display (VAD) wind, and radar radial velocity (VR) observations are assimilated. The pseudo-$w$ data is also assimilated for the DA-W experiments.

### 4.2.2 Results

To statistically evaluate the performance of the CTRL and DA-W experiments for precipitation forecasting, the equitable threat score (ETS; Gandin and Murphy, 1992), the neighborhood-based fractions skill score (FSS; Roberts and Lean, 2008), and the bias score (BIAS; Anthes, 1983) are calculated for the forecast hourly accumulated precipitation. Forecasts with higher ETS (close to 1) and FSS (close to 1) and closer BIAS to 1, demonstrate better forecast skills. Figs. 4-6 present the 10-day averaged forecast skills for the hourly accumulated precipitation from 0600 UTC to 1200 UTC. For the threshold of 1 mm h$^{-1}$, it is not always the case that the ETS improves as the number of assimilation times increases for both the CTRL and DA-W experiments. However, with an increase in the scoring threshold, especially for 20 mm h$^{-1}$, a higher score is generally achieved with more assimilation times, indicating a positive impact of multiple assimilations on the forecast. When comparing the ETS scores of the CTRL and DA-W experiments with the same assimilation times, it can be seen that the DA-W experiment has a neutral or slightly negative effect on the forecast at the threshold of 1 mm h$^{-1}$. However, at thresholds of 5 and 20 mm h$^{-1}$, the DA-W experiment achieves higher scores than the CTRL experiment in most situations, regardless of whether it involves multiple or single assimilation. Moreover, the experiment with 3 assimilation times (denoted by experimental names ending with "3CY") demonstrates the most significant improvements compared to the experiments with other assimilation times.

The FSS scores provide clearer results: for experiments with the same assimilation times in CTRL and DA-W (e.g., the DA-W-2CY compared to the CTRL-2CY experiment), the DA-W experiment consistently achieves better scores, indicating that the assimilation of $w$ has a positive impact on the forecast of precipitation location. From the BIAS scores, the DA-W experiments have a neutral impact on the forecast compared to the CTRL experiments. In the first 3-hour forecast, the DA-W experiment generally performs worse than the CTRL experiment (with the same assimilation times) for each threshold value, primarily because it produces more false alarms. However, in the latter 3-hour forecast, the DA-W experiment demonstrates better scores compared to the CTRL experiment.

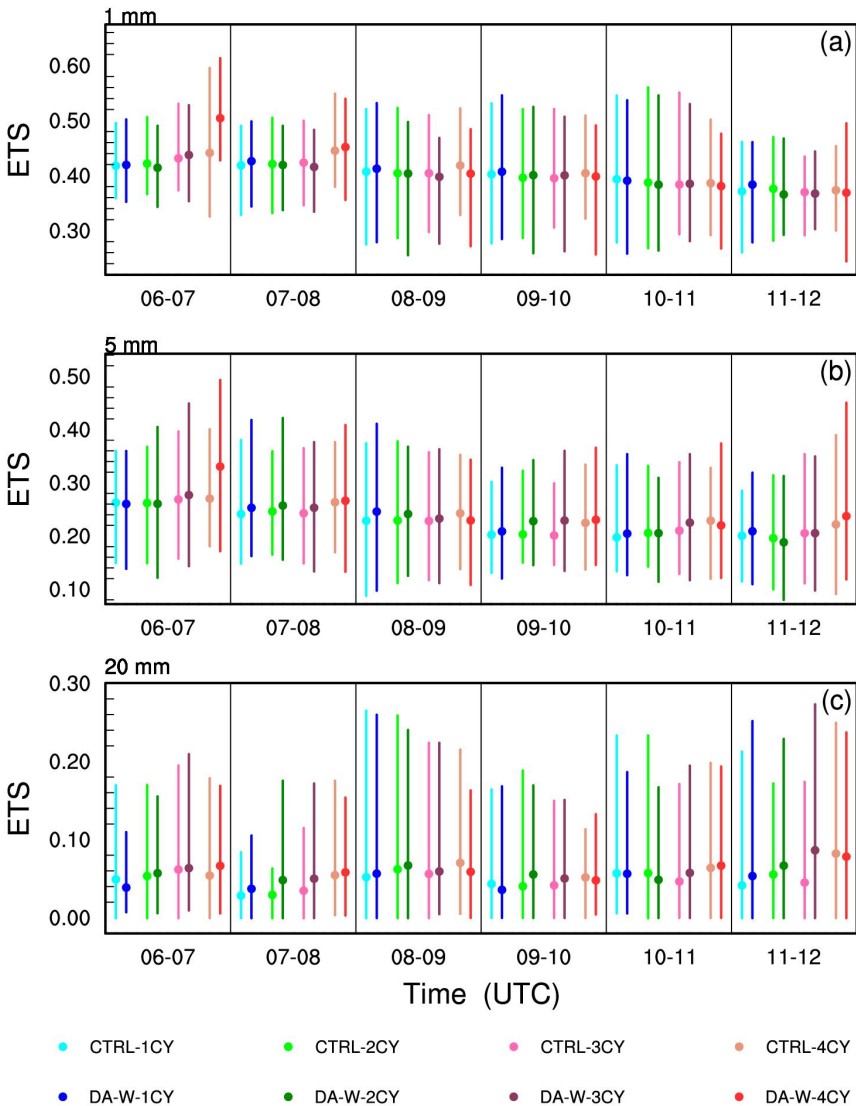

CTRL-1CY •    CTRL-2CY •    CTRL-3CY •    CTRL-4CY •

DA-W-1CY •    DA-W-2CY •    DA-W-3CY •    DA-W-4CY •

**Figure 4: The 10-d (July 1 to July 10, 2020) averaged equitable threat score (ETS; solid dots) of the predicted hourly accumulated precipitation from 0600 to 1200 UTC of the CTRL and DA-W experiments for thresholds of (a) 1 mm h[-1], (b) 5 mm h[-1], and (c) 20 mm h[-1]. The top (bottom) of the line that passes through the solid dot corresponds to the maximum (minimum) ETS value for those 10 days.**

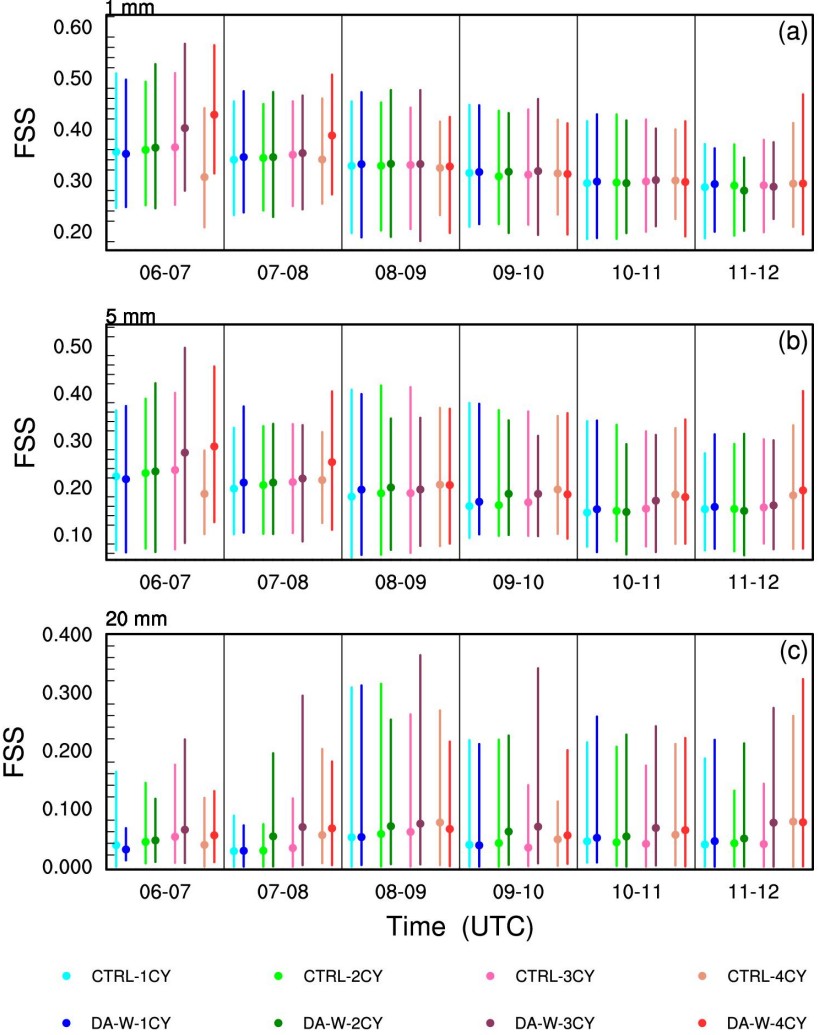

**Figure 5: Same as Fig. 4 but for the neighborhood-based fractions skill score (FSS).**

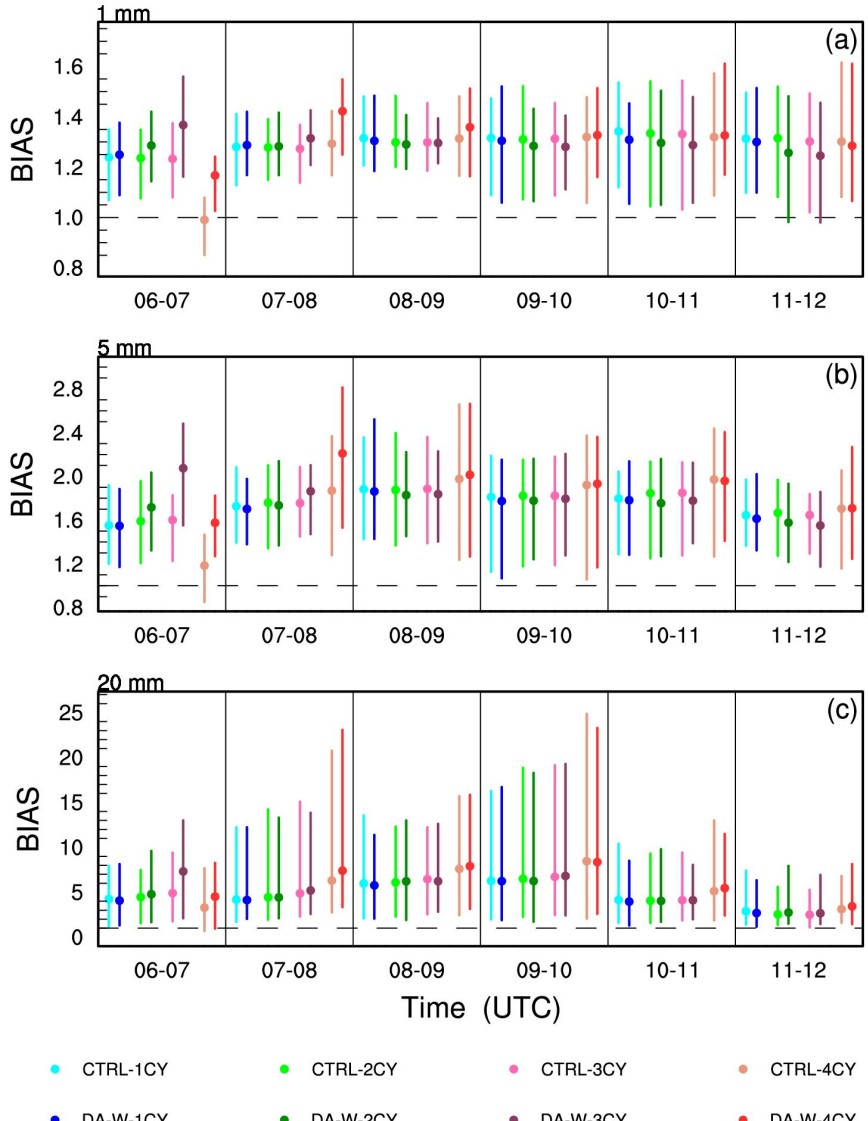

**Figure 6: Same as Fig. 4 but for the bias score (BIAS). The black dashed line represents BIAS value equals to 1.**

### 4.3 Case study

255 The case initialized on July 9, 2020 from the batch experiments is chosen to have a further test. Fig. 7 presents the ETS, FSS, and BIAS scores for different thresholds. In both the CTRL and DA-W experiments, increasing the assimilation times does not necessarily result in higher ETS scores, particularly for the 1 mm h$^{-1}$ threshold. However, when comparing the CTRL and DA-W experiments with the same assimilation times, the DA-W experiment consistently achieves higher scores. The FSS scores indicate that, except for the period from 0600 to 0700 UTC, the hourly accumulated precipitation exhibits higher

scores with more assimilation times, and the DA-W experiment consistently outperforms the CTRL experiment. Regarding the BIAS scores, the DA-W experiment has a neutral effect on the forecast compared to the CTRL experiment.

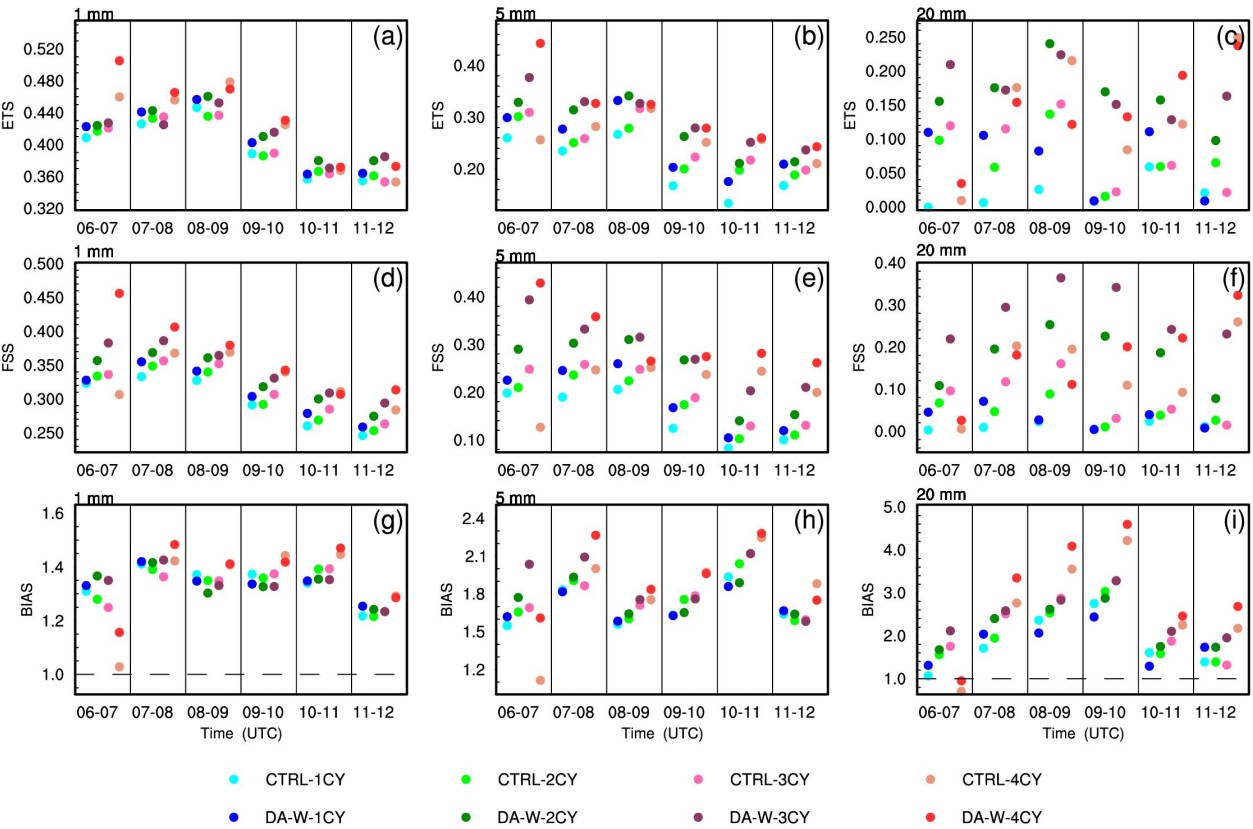

Figure 7: The equitable threat score (ETS; (a)–(c)), the neighborhood-based fractions skill score (FSS; (d)–(f)), and the bias score (BIAS; (g)–(i)) for the predicted hourly accumulated precipitation of the CTRL and DA-W experiments (the black dashed lines in
(g) and (i) represent BIAS values equal to 1). The analysis focuses on thresholds of 1 mm h⁻¹, 5 mm h⁻¹, and 20 mm h⁻¹ for the case initialized at 0000 UTC on July 9, 2020.

Fig. 8 displays the 6-hour accumulated precipitation of the CTRL-4CY and DA-W-4CY experiments, with the majority of precipitation occurring in Jiangxi Province. The center of heavy precipitation exhibits a maximum 6-hour accumulated precipitation exceeding 100 mm. The CTRL-4CY experiment successfully captures the forecast location of this area of
heavy rainfall, although the overall intensity of the precipitation is low. In contrast, the DA-W-4CY experiment performs better in forecasting the intensity of heavy precipitation.

In Fig. 8(a), line A-B represents the observed main precipitation belt. Fig. 9 shows the sections along line A-B for the CTRL-4CY and DA-W-4CY experiments at 0700 UTC on July 9, 2020. The DA-W-4CY experiment effectively enhances the $w$ values across the entire model layers. This enhancement is achieved by generating increments of wind convergence
(less than $-4\times10^{-4}$ s⁻¹) at the lower (the 13th) level of the model while inducing divergence or weak wind convergence increments at the middle (the 23rd) level of the model (Fig. 10). This configuration of the horizontal wind field enables the

model to generate specific vertical velocities in the middle and lower levels, leading to a decrease in water vapor below 850 hPa compared to the CTRL-4CY experiment (Fig. 9(c)). Simultaneously, positive increments of water vapor are observed in the middle and upper layers of the model. Consequently, upward movement enhances the vertical transport of water vapor, promoting water vapor saturation and facilitating cloud formation, ultimately resulting in rainfall.

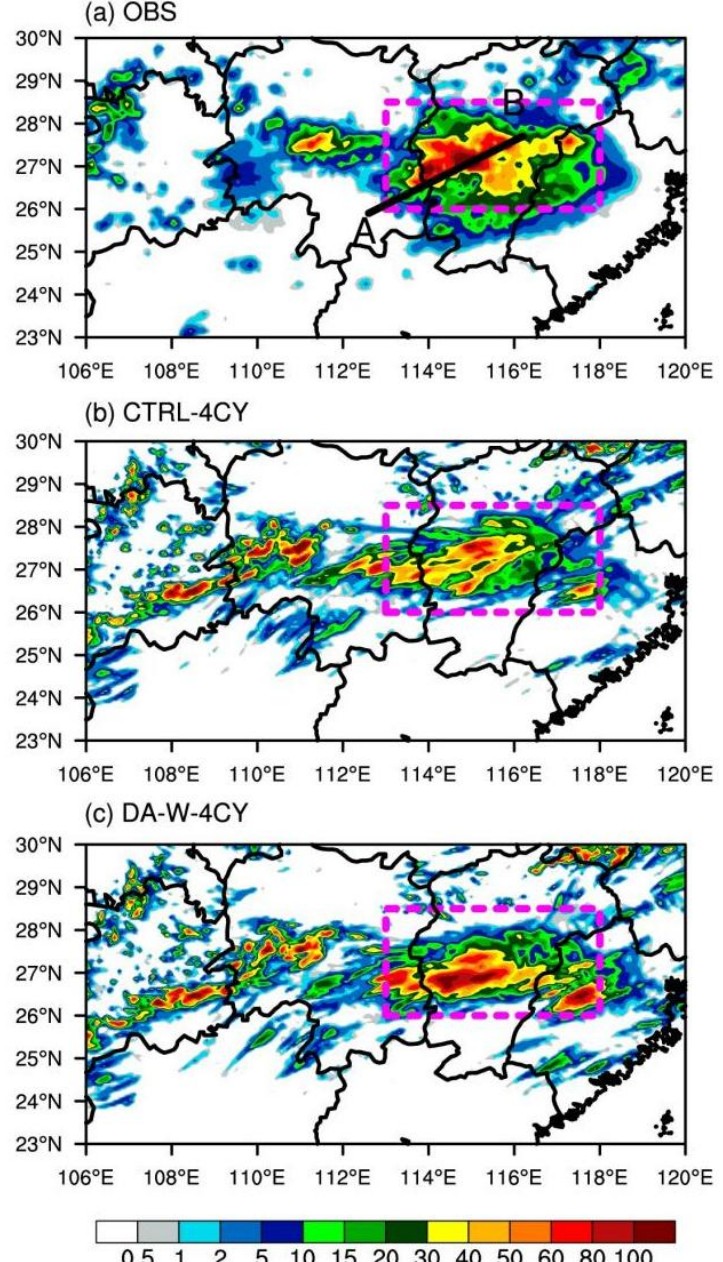

**Figure 8: The 6-hour (0600 to 1200 UTC) accumulated precipitation (units: mm) on July 9, 2020 for the (a) observations (OBS), (b) CTRL-4CY, and (c) DA-W-4CY experiments. The areas enclosed by dotted purple lines indicate regions with observed strong rainfall.**

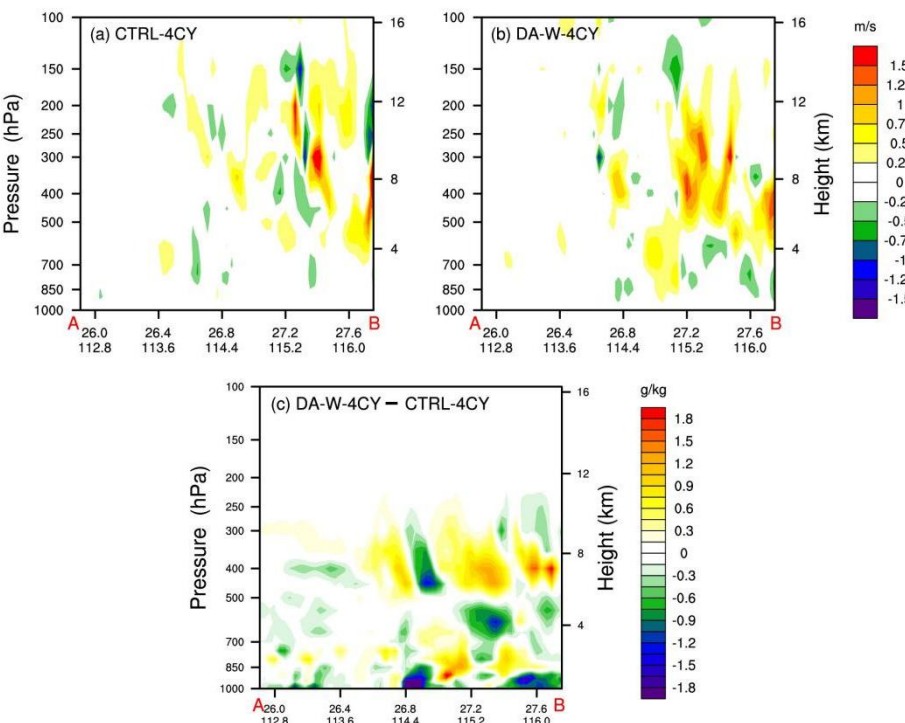

**Figure 9: Cross sections of the $w$ (units: m s$^{-1}$) at 0700 UTC on July 9, 2020, along line A–B in Fig. 8 (a) for the (a) CTRL-4CY and (b) DA-W-4CY experiments. (c) represents the difference in water vapor between the CTRL-4CY and DA-W-4CY experiments (units: g kg$^{-1}$).**

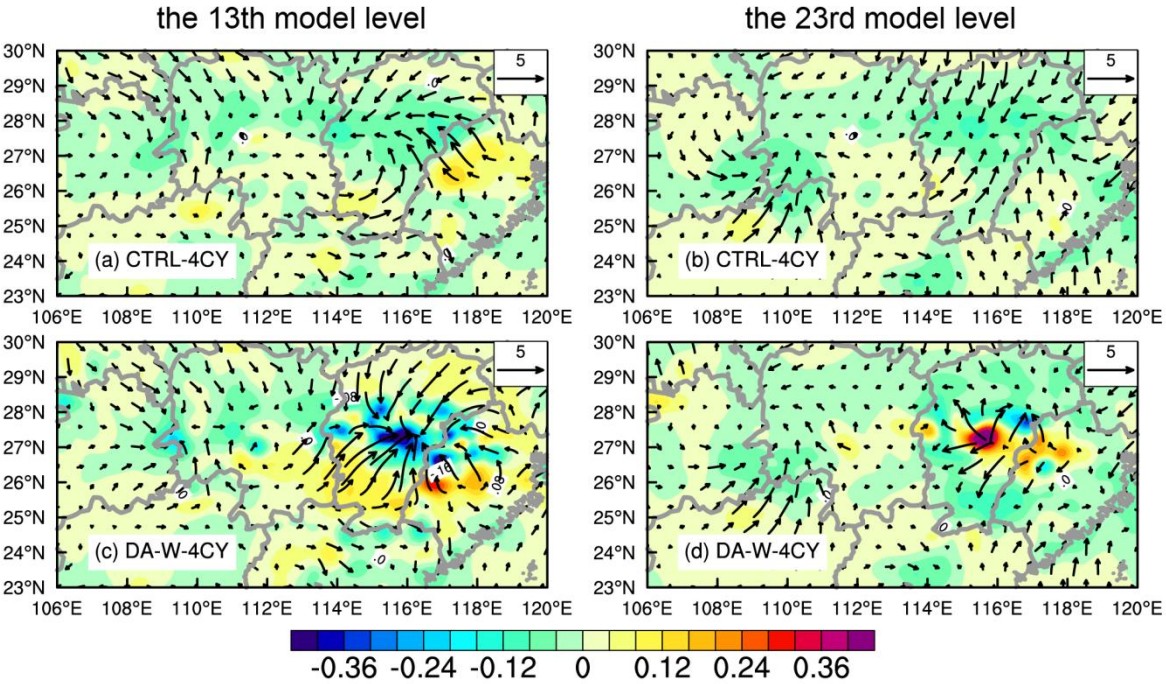

Figure 10: Analysis increments of horizontal wind (vector; unit: m s$^{-1}$) and horizontal wind divergence (color; unit: $10^{-4}$ s$^{-1}$) of the (a, b) CTRL-4CY and (c, d) DA-W-4CY experiments at the (a, c) 13th (~850 hPa) and (b, d) 23rd (~500 hPa) model levels at 0600 UTC on July 9, 2020.

## 5 Conclusions and discussion

Dynamical processes, especially vertical air motions, play a crucial role in convective precipitation forecasts as they contribute to the development of clouds and precipitation. In this study, a 3D-Var data assimilation scheme for $w$, based on the adiabatic Richardson equation, is developed within the high-resolution (3 km) CMA-MESO model. The CMA-MESO 3D-Var system employs the horizontal wind components $u$ and $v$ as momentum control variables. The observation operator for $w$ establishes the relationship between $w$ and $u$, $v$, as well as $\Pi$ (dimensionless air pressure). This allows the $u$ and $v$ fields to be updated directly by assimilating $w$ observations. The results of the single observation test indicate a reasonable distribution of horizontal wind increments. Specifically, horizontal wind convergence (resulting from the assimilation of a positive value of $w$) is observed at the lower model level (~850 hPa), while a horizontal wind divergence tends to occur at the higher model level (~400 hPa). These adjustments contribute to the establishment or reinforcement of convection in these areas.

The impact of assimilating pseudo-observations of $w$ on forecasts is then investigated through the study of a series of continuous 10-day runs and an individual case within it. The pseudo-$w$ observations are derived based on the empirical relationship between radar reflectivity factor and $w$. It should be noted that the $w$ assimilation scheme established in this study is also applicable to other sources of $w$. Two sets of experiments were configured, including CTRL and DA-W

experiments with different assimilation iterations. Both sets of experiments assimilated aircraft measurements, radiosondes, and other observations (for a comprehensive list, refer to Fig. 3 (b)) at 1-hour intervals during a 3-hour data assimilation period. In addition, the pseudo-$w$ observations are also assimilated in the DA-W experiments. The DA-W experiment achieves better FSS scores than the CTRL experiment (with the same assimilation times), indicating an improved positional forecast accuracy of precipitation. As for the ETS skills, the DA-W experiment demonstrates improved performance compared to the CTRL experiment at higher thresholds (5 and 20 mm h$^{-1}$). However, the DA-W experiments tend to generate some spurious precipitation, resulting in inconsistent improvements in BIAS compared to those of the CTRL experiments. The individual case study indicates that the DA-W experiment contributes to enhancing upward motion in convective regions, resulting in improved forecasts of heavy precipitation that are closer to the observations.

Our study has successfully achieved direct assimilation of $w$ within the current CMA-MESO 3D-Var system, yielding promising preliminary results. However, there are certain limitations that cannot be overlooked and require further attention. For instance, 1) The adjustments in temperature and humidity increments are achieved by weak physical constraints, and it would be better to take into account the multivariate correlation between control variables (Hollingsworth and Lönnberg, 1986; Barker et al., 2004). 2) The pseudo-observations of $w$ used in this study are derived from radar reflectivity. However, certain instruments, such as wind profilers, are capable of acquiring $w$ observations. In addition, the radial velocity also includes vertical velocity information. It is valuable to conduct further testing to assess the impact of assimilating $w$ from these observations on forecasts. 3) Radar reflectivity observations have traditionally been employed to initialize the moisture field and hydrometeors of regional models (e.g., Albers et al., 1996; Sun and Crook, 1997; Hu et al., 2006; Wang et al., 2013; Lai et al., 2019; Liu et al., 2022), and the benefits of assimilating high-resolution radar data might diminish due to inconsistencies in dynamic information. It is imperative to concurrently update the dynamical variables to maintain a balanced initial field. Our approach could potentially address this issue, since direct assimilation of $w$ observations is possible. As the CMA-MESO model progresses in incorporating radar reflectivity factor assimilation, the combined assimilation of water vapor, hydrometeors, and $w$ warrants further exploration.

**Code availability**

The CMA-MESO v5.0 source code is provided by the Chinese Meteorological Administration and cannot be publicly available due to the copyright license requirement from the China Meteorological Administration Earth System Modeling and Prediction Centre (CEMC). If someone wishes to acquire the code to reproduce the study, please contact the operational
management department of the CEMC via email (sunqin@cma.gov.cn) or phone (+86-10-58994128). The code of the observation operator, the tangent linear of the observation operator, and the adjoint operator for $w$ is available at https://doi.org/10.5281/zenodo.10073822.

**Data availability**

The model outputs from the "Single-point observation experiment" and "Case study" sections, along with the processed data
from the "The batch experiment" section of the paper, are available at https://doi.org/10.5281/zenodo.10867909. The NCEP GFS data used are available at https://rda.ucar.edu/datasets/ds084.1/. The raw Doppler radar and precipitation observations are provided by the Chinese Meteorological Administration and can be obtained via request from http://www.cma.gov.cn/en2014/.

**Author Contributions**

Yi Yang conceived the idea and designed the research. Hong Li performed the research and wrote the first draft of the manuscript. All authors discussed the results and contributed to writing and revisions.

**Conflict of interest**

The authors declare no conflicts of interest or competing financial interests.

**Acknowledgments**

The authors gratefully acknowledge the CEMC for providing the source code of CMA-MESO v5.0. Additionally, the authors appreciate the NCEP for providing the GFS data and the Chinese Meteorological Administration for providing the radar and precipitation data. Furthermore, special thanks to the Supercomputing Center of Lanzhou University for computational support.

**Financial support**

This study was jointly supported by the Youth Science and Technology Foundation Program of Gansu Province (23JRRA1327), the National Natural Science Foundation of China (42205050), and the Gansu Provincial Association of Science and Technology Innovation Drive Promotion Project (GXH20230817-7).

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
