# Peer review of "A 3D-Var Assimilation Scheme for Vertical Velocity with the CMA-MESO v5.0"

_Geoscientific Model Development, 2023_

## Referee Comment (RC2)

The article presents a 3DVAR assimilation scheme for w, which appears reasonable and has a positive impact based on results from a heavy-rain event and a 10-day batch experiment. However, there are some doubts:

1.From the batch experiment results, it was found that w assimilation has a small impact on the forecast. However, in the case study where w was assimilated in a cyclical manner, it led to significant improvements in the forecast. It is suggested that the authors also set multiple w assimilations in the batch experiment section.

2.As mentioned by the author, some studies assimilate w from total lightning data. In this article, however, the w observations are derived from radar reflectivity data. It is unclear how the authors obtained at the approximate magnitude of the w values using this method. In addition, were the authors able to compare the radar reflectivity-derived w with the w of the model background field to determine any differences in magnitude between the two values? If there is a significant difference, it may be necessary to remove the larger w values during the assimilation process.

Some minor revisions are as follows:
Page 1:
Line 21: Change "the result indicates" to "the results indicate".
Line 22: The statement "leading to improved equitable threat score (frequency skill score) for the first 1 h (3 h) precipitation forecasts" may cause confusion, please describe it in detail.
Line 23: Change "assimilated" to "assimilation".

Page 2:
Line 36: Change "allows they" to "allows them".
Line 41-42: Delete "of vertical velocity".
Line 48: Add "s" to the word "field".

Page 3:
Line 75: Delete "real".
Lines 89-91: This sentence is quite difficult to understand. I suggest that it be described simply and clearly.
Line 92: Delete "to assimilate w observation directly".
Line 96: Add "s" to the word "adjust".

Page 14:
Line 279: Change "wish" to "wishes".

---

## Author Comment (AC1)

**Response to reviewers of the manuscript**

"*A 3D-Var Assimilation Scheme for Vertical Velocity with the CMA-MESO v5.0*"

H. Li, Y. Yang, J. Sun, Y. Jiang, R. Gan, and Q. Xie

for Geoscientific Model Development

**Response to Reviewer #1**

Reviewer #1

In this study, a vertical velocity (w) assimilation scheme is developed in the 3DVar method. In this w assimilation scheme, the Richardson equation is employed as the operator for w, enabling the 3DVar system to update dynamic and mass variables from w assimilation. The w assimilation scheme looks reasonable to me. However, I have concerns about the experimental configurations for both the single case study and batch experiments in the manuscript. Therefore, I recommend a major revision.

**Response:** We gratefully thank the reviewer for your time spent making the constructive remarks and useful suggestions, which has significantly raised the quality of the manuscript and has allowed us to improve the manuscript. Each suggestion has been accurately incorporated and addressed in the revised manuscript. Please see our detailed response below in blue.

Major comments:

1.  The authors highlight the use of the Richardson equation as the operator for $w$ to reduce imbalance, but there is a lack of corresponding discussion and figures to support this assertion. I strongly recommend the authors incorporate related discussions and provide figures that illustrate the effectiveness of employing the Richardson equation for $w$ in reducing imbalance.

**Response:** We appreciate the reviewer's valuable suggestion. The authors here expressed that using the Richardson equation as an observation operator for assimilating $w$ to reduce imbalance is in comparison to the direct assimilation of $w$ by expanding it as a control variable (the observation operator is simplified as an interpolation operator from model space to observation space). Unfortunately, $w$ is not treated as a control variable in the current 3D-Var system of CMA-MESO. Therefore, the authors regret that, at present, we are unable to conduct further experimental validation and can only rely on the algorithm's design to explain this issue. Using $w$ as an extended momentum control variable may introduce excessive noise, as $w$ only has an impact on itself after assimilation, unless a physical constraint is incorporated through a physical transformation in the background error or a physical weak constraint is added to the cost function. To avoid this problem, Chen et al. (2020) assimilated the pseudo-horizontal wind convergence derived from the $w$ fields using the mass continuity equation. In our study, the adiabatic Richardson equation is employed as the observation operator for $w$. This observation

operator combines the continuity equation, adiabatic equations, and hydrostatic relation. As a result, the update of other variables is physically constrained when $w$ observations are assimilated.

Nevertheless, the authors attempted to analyze the correlation between $w$ and other variables (including the zonal and meridional winds $u$ and $v$, temperature $t$, and water vapor mixing ratio $q_v$) based on 120 forecast samples. Fig. R1 shows the correlation coefficients between the time series of variable $w$ and the other model variables at each point on the 700 hPa level. It can be seen that the correlation between variable $w$ and the other variables is complex. In Fig. R2, the correlation coefficients between variable $w$ at the point (32.5°N, 112.51°E) and the other model variables at different points show that there are some spurious correlations exist at distant distances.

Considering the non-hydrostatic characteristics of meso-microscale convective systems, the authors plan to conduct non-hydrostatic assimilation experiments for $w$ in which $w$ is expanded as a control variable in our upcoming work. This will involve re-estimating the background error covariance and allow for further investigate the impact of assimilating $w$ in this manner on the update of other model variables.

[Figure]

**Figure R1** The spatial distribution of correlation coefficients between variable $w$ and (a) zonal wind $u$, (b) meridional wind $v$, (c) temperature $t$, and (d) water vapour

mixing ratio $q_v$ at 700 hPa. These statistics are based on 120 forecast samples.

[Figure]

**Figure R2** The spatial distribution of correlation coefficients between the time series of variable $w$ at the point (32.5°N, 112.51°E) and the time series of (a) zonal wind $u$, (b) meridional wind $v$, (c) temperature $t$, and (d) water vapour mixing ratio $q_v$ at other model points on the 700 hPa level.

2. In the single observation test, only horizontal increments are presented, while the *w* observation captures vertical motions. It would be more informative to display vertical cross-sections of the increments, revealing the vertical spread of assimilating *w* single observations.

**Response:** Thank you for your suggestion. Following the reviewer's suggestion, the vertical cross-sections of the analysis increment have been included as Fig. 2 in the revised manuscript. Additionally, the following statement has been added (Lines 148–152 in the revised manuscript): "From the vertical cross-section of the analysis increment for each variable (Fig. 2), it can be seen that the increase in specific humidity is primarily concentrated in the lower layer below the observation location, while the increases in the other three variables are distributed throughout the entire layer. Regarding the increase in horizontal wind $u$, below the single point observation, there is a convergence of $u$ wind that extends to the ground. Above the single point observation, there is a divergence of $u$ wind that extends to approximately 150 hPa".

[Figure]

**Figure 2 of the revised manuscript**. The analysis increments of (a) zonal wind $u$, (b) meridional wind $v$ (unit: m s$^{-1}$), (c) specific humidity (unit: kg kg$^{-1}$), and (d) temperature (unit: K) in a vertical cross-section at 38.0 °N at 1500 UTC on July 4, 2020, for the single observation experiment. The solid black dot in the figure represents the location of the single observation.

I am particularly interested in understanding the vertical range of assimilating w. Does it extend from the bottom level to the top level? Are there any constraints limiting the impacts of w assimilation?

**Response:** Yes, the assimilation of $w$ extends from the bottom level to the top level. The current design does not impose any constraints on the impacts of $w$ assimilation in the vertical direction, although it is necessary to set limits, particularly for unreasonable increments at higher levels of the model. However, a specific method for rationalizing the vertical constraint has not been devised yet, except for a crude approach that restricts impacts below a certain model layer. Exploring more reasonable ways to limit the impact of $w$ assimilation in the vertical direction will be part of the author's future work.

Additionally, based on Fig. 1, the range of significant increments appears to be approximately 8 degrees. Have you considered tuning the decorrelation scales of BEC for *w* assimilation in real case experiments? I believe adjusting these

scales could be crucial for constraining the impact radii of $w$ observations in convective-scale data assimilation.

**Response:** Yes, regarding the optimal decorrelation scales for static background error covariances, the authors conducted several sensitivity experiments for the real case study. Specifically, the length scales of 50 km are employed for horizontal wind components based on the study conducted by Xiao et al. (2020), which utilized the length scales of 48 km for horizontal wind components. However, from Fig. R3, there is a slight difference in the forecast composite radar reflectivity between the control (CTRL) and DA-W ( $w$ pseudo-observations are cyclically assimilated with an interval of 1 h from 1300 to 1500 UTC on July 4, 2020) experiments. In the study, a relatively large scale factor of 400 km was used for several reasons. First, the derived $w$ pseudo-observations are mostly small values (shown in Fig. R4 (c)–(c3)), with the majority (90.4%) falling within the range of 0.0-0.6 m/s. Second, the observation operator for $w$, the adiabatic Richardson equation, combines the continuity equation, adiabatic equations, and the hydrostatic relation, making it more suitable for large-scale systems.

[Figure]

**Figure R3** The observed (OBS; (a1)–(c1)) composite radar reflectivity (units: dBZ) and the forecast composite radar reflectivity of the (a2)–(c2) CTRL and (a3)–(c3)

[Figure]

DA-W experiments at (a1)–(a3) 1600 UTC, (b1)–(b3) 1700 UTC, and (c1)–(c3) 1800 UTC on 4 July, 2020.

**Figure R4** The observed composite radar reflectivity greater than 35 dBZ ((a)–(a3); units: dBZ), the derived maximum $w$ pseudo-observations ((b)–(b3); units: m s$^{-1}$), and the frequency distribution of $w$ pseudo-observations ((c)–(c3)) at (a)–(c) 1200 UTC, (a1)–(c1) 1300 UTC, (a2)–(c2) 1400 UTC, and (a3)–(c3) 1500 UTC on July 4 2020.

3. The assimilation configurations in both the heavy rainfall case study and batch experiments appear unconventional. In the case study, CNTL performs no assimilation, while DA-W assimilates $w$ hourly for only three hours. This leads to DA-W being initialized four times additionally, potentially introducing significant impacts. It's challenging to attribute these impacts solely to the assimilation of $w$. In the batch experiments, both cases seem to involve single-time assimilation each day, likely contributing to minimal

impacts from *w* assimilation. I recommend the authors rerun both batch and single-case experiments, considering an increased number of assimilation cycles per day. For consistency, the single case experiments could from one day of the batch experiments.

**Response**: Thank you for your valuable comments. Based on your suggestion, we have conducted a new set of batch experiments. In Figure 3 (b) of the revised manuscript, both the CTRL and DA-W experiments were initialized at 0000 UTC daily from July 1 to July 10, 2020, and run until 1200 UTC each day. The first 3 hours were considered as a "spin-up" period. In the CTRL experiments, observations from aircraft measurements, radiosondes, and other sources (for a comprehensive list, refer to Fig. 3 (b)) were assimilated from 0300 to 0600 UTC with a 1-hour assimilation interval (radial velocity observations are available at each analysis time, while other data sources are only available at 0300 and 0600 UTC). The CTRL-1CY experiment indicates assimilation at 0300 UTC only, while the CTRL-2CY experiment represents assimilation at 0300 and 0400 UTC, and so on (the number preceding the experiment name "CY" represents the assimilation iterations). The DA-W experiments are similar to the CTRL experiments, but include the assimilation of *w* pseudo-observations (*w* pseudo-observations are available at each analysis time).

[Figure]

**Figure 3 (b) of the revised manuscript.** Illustration of the numerical experimental scheme for the CTRL and DA-W batch experiments. Both experiments utilize NCEP GFS data as the initial condition (IC) and boundary condition (BC). The abbreviation "Fcst" represents forecast. The assimilated data comprises conventional observations from aircraft measurements (AIREP), radiosondes (TEMP), ships (SHIP), and ground stations (SYNOP). In addition, cloud-track-wind (SATOB), precipitable water derived from the Global Positioning System (GPSPW), refractivity radio-occultation data

from the Global Navigation Satellite System (GNSS-RO), wind profiler radar (WPR), velocity-azimuth display (VAD) wind, and the radar radial velocity (VR) are assimilated. The pseudo-$w$ data is also assimilated for the DA-W experiments.

Figs. 4-6 in the revised manuscript present the 10-day averaged forecast skills for hourly accumulated precipitation from 0600 UTC to 1200 UTC. For the threshold of 1 mm h$^{-1}$, it is not always the case that the ETS score improves as the number of assimilation times increases for both CTRL and DA-W experiments. However, with an increase in the scoring threshold, especially for 20 mm h$^{-1}$, a higher score is generally achieved with more assimilation times, indicating a positive impact of multiple assimilation on the forecast. However, with an increase in the scoring threshold, especially for 20 mm h$^{-1}$, a higher score is generally achieved with more assimilation times, indicating a positive impact of multiple assimilation on the forecast. When comparing the ETS scores of the CTRL and DA-W experiments with the same assimilation times, it can be seen that the DA-W experiment has a neutral or slightly worse effect on the forecast at threshold of 1 mm h$^{-1}$ compared to the CTRL experiment. However, at thresholds of 5 and 20 mm h$^{-1}$, the DA-W experiment achieves higher scores than the CTRL experiment in most situations, regardless of multiple or single assimilation. Moreover, the experiment with 3 assimilation times (denoted by experiment names ending with "3CY") demonstrate the most significant improvements compared to experiments with other assimilation times.

The FSS scores provide clearer results: for experiments with the same assimilation times in CTRL and DA-W (e.g., DA-W-2CY compared to CTRL-2CY experiment), the DA-W experiment consistently achieves better scores, indicating that the assimilation of $w$ has a better adjustment effect on the forecast of precipitation location. From the BIAS scores, the DA-W experiments have a neutral impact on the forecast compared to the CTRL experiments. In the first 3-hour forecast, the DA-W experiment generally performs worse than the CTRL experiment (with the same assimilation times) for each threshold value, primarily due to producing more false alarms. However, in the latter 3-hour forecast, the DA-W experiment demonstrates better scores compared to the CTRL experiment.

[Figure]

**Figure 4 of the revised manuscript.** The 10-d (July 1 to July 10, 2020) averaged equitable threat score (ETS; solid dots) of the predicted hourly accumulated precipitation from 0600-1200 UTC of the CTRL and DA-W experiments for thresholds of (a) 1 mm h⁻¹, (b) 5 mm h⁻¹, and (c) 20 mm h⁻¹. The top (bottom) of the line that passes through the solid dot corresponds to the maximum (minimum) ETS value for those 10 days.

[Figure]

**Figure 5 of the revised manuscript.** Same as Fig. 4 but for the neighborhood-based fractions skill score (FSS).

[Figure]

**Figure 6 of the revised manuscript.** Same as Fig. 4 but for the bias score (BIAS). The black dashed line represents BIAS value equals 1.

Following the reviewer's suggestion, we have replaced the individual test with a case in the batch experiments, specifically the case initialized on July 9, 2020. Fig. 7 presents the ETS, FSS, and BIAS scores for different thresholds. In both the CTRL and DA-W experiments, increasing the assimilation times does not necessarily result in higher ETS scores, particularly for the 1 mm h$^{-1}$ threshold. However, when comparing the CTRL and DA-W experiments with the same assimilation times, the DA-W experiment consistently achieves higher scores. The FSS scores indicate that, except for the period from 0600 to 0700 UTC, the hourly accumulated precipitation exhibits higher scores with more assimilation times, and the DA-W experiment consistently outperforms the CTRL experiment. Regarding the BIAS scores, the DA-W experiment has a neutral effect on the forecast compared to the CTRL experiment.

Fig. 8 displays the 6-hour accumulated precipitation of the CTRL-4CY and DA-W-4CY experiments, with the majority of precipitation occurring in Jiangxi

Province. The heavy precipitation center exhibits a maximum 6-hour accumulated precipitation exceeding 100 mm. The CTRL-4CY experiment successfully captures the forecast location of this heavy rainfall area, although the overall precipitation intensity is low. In contrast, the DA-W-4CY experiment performs better in forecasting the intensity of heavy precipitation.

In Fig. 8 (a), line A-B represents the observed main precipitation belt. Fig. 9 shows the sections along line A-B for the CTRL-4CY and DA-W-4CY experiments at 0700 UTC on July 9, 2020. The DA-W-4CY experiment effectively enhances the $w$ values across the entire model layers, resulting in a negative water vapor increment below the model's 850 hPa compared to the CTRL-4CY experiment. Simultaneously, positive increments of water vapor are observed in the middle and upper layers of the model.

[Figure]

**Figure 7 of the revised manuscript.** The equitable threat score (ETS; (a)–(c)), the neighborhood-based fractions skill score (FSS; (d)–(f)), and the bias score (BIAS; (g)–(i)) for the predicted hourly accumulated precipitation of the CTRL and DA-W experiments (the black dashed lines in (g) and (i) represent BIAS value equals 1). The analysis focuses on thresholds of 1 mm h⁻¹, 5 mm h⁻¹, and 20 mm h⁻¹ for the case initialized at 0000 UTC on July 9, 2020.

[Figure]

**Figure 8 of the revised manuscript.** The 6-hour (0600-1200 UTC) accumulated precipitation (units: mm) on July 9, 2020 for (a) observations (OBS), (b) CTRL-4CY, and (c) DA-W-4CY experiments. The areas enclosed by dotted purple lines indicate regions with observed strong rainfall.

[Figure]

**Figure 9 of the revised manuscript.** Cross sections of the $w$ (units: m s$^{-1}$) at 0700 UTC on July 9, 2020, along line A–B in Fig. 8 (a) for the (a) CTRL-4CY and (b) DA-W-4CY experiments. (c) represents the difference in water vapor between the CTRL-4CY and DA-W-4CY experiments (units: g kg$^{-1}$).

Minor comments:

Line 42: Change "vertical velocity" to "w". Check it throughout the manuscript.

**Response:** Thank you for your suggestion. We have revised it as the reviewer suggested throughout the manuscript.

Line 54: Change "computing" to "computational".

**Response:** Thank you for your suggestion. It has been revised as suggested (Line 57 in the revised manuscript).

Line 54: The physical constraint is implicitly considered in the ensemble BEC in the EnKF-based methods.

**Response:** Yes, we agree. Thank you for your suggestion, and we apologize for our less rigorous statement. The statement "2) lack of strict physical constraints except for the four-dimensional variational method" has been revised to "2) lack of strict physical constraints" (Line 58 in the revised manuscript).

Line 58: the observation operator of w is not the problem. The issue is the effectiveness of assimilating w in the forecasts.

**Response:** Thank you for your careful consideration. Following the reviewer's suggestion, we have revised the sentence as "Within the 3D-Var framework,

assimilating $w$ faces numerous challenges, the most significant of which is the development of an effective assimilation method that produces a reasonable positive impact on forecasts" to improve its clarity and appropriateness (Lines 62–63 in the revised manuscript).

Line 86: "wind" should be "winds".

**Respon**se: Thank you for your suggestion. It has been revised as suggested (Line 86 in the revised manuscript).

Line 91: It is not accuracy. H links the model state variables to the observed variables, not the control variables.

**Response:** We apologize for this mistake. The "control variables" has been revised to "the model state variables" in the revised manuscript (Line 94).

Line 96-97: This sentence is not accuracy enough. The observation operator combines the dynamic and mass fields, but it cannot adjust these variables. Maybe it could be modified to "enabling the 3DVar method to adjust ….".

**Response:** Thank you for your suggestion. Yes, our previous statement was not precise enough. According to the reviewer's suggestion, we have revised the sentence to "The Richardson equation combines the continuity equation, adiabatic equations, and hydrostatic relation, which enables the 3D-Var method to adjust the dynamic and mass fields simultaneously and result in a more balanced analysis field." (Lines 99–100 in the revised manuscript).

Line 100: add "height" after "top".

**Response:** Thank you for your suggestion. Added as suggested (Line 103 in the revised manuscript).

Line 116-119: Make it simple. Maybe the (2) and (3) can be combined.

**Response:** Thank you for your suggestion. The statement does cause tediousness here, and has been revised to "the tangent linear of the observation operator and its adjoint for the $w$ term are included to calculate the cost function and its gradient values" as the reviewer suggested (Line 119–120 in the revised manuscript).

Line 122: It is common to use bold H to represent the tangent linear observation operator and with the subscript T to represent its adjoint operator.

**Response:** Thank you for your suggestion. The symbols $H'$ and $H^T$ have been revised to $\mathbf{H}$ and $\mathbf{H}^T$, respectively, to represent the tangent linear observation operator and its adjoint operator throughout the entire revised manuscript.

Eq. (9): The cost function J in Eq. (1) corresponds to analysis state x, while in Eq. (9) it is for the control variable. Modify it to make it consistent.

**Response:** Following the review's suggestion, the cost function $J$ in Eq. (1) has been revised to $J(c_v)$ to maintain consistency with Eq. (9) (Eq. (10) in the revised manuscript).

Fig. 3: How is the bias score calculated? Is it the frequency bias? For me, the values of bias are too large, indicating a significant overestimation of both experiments.

**Response:** We appreciate the reviewer's careful attention, and we apologize for any confusion caused. The BIAS score, which represents the frequency bias, is calculated using the following equation:

$$\text{BIAS} = \frac{\text{hits+false alarms}}{\text{hits+misses}}, \quad (R1)$$

where the indicator "hits" refers the event forecast to occur and did occur; "false alarms" refers to the event forecast to occur, but did not occur; "misses" refers to the event forecast not to occur, but did occur. The specific calculation for each indicator was performed using the neighborhood-based method described in reference Clark et al. (2010). "false alarms" are assigned when an event is forecast at a grid point and not observed within a neighborhood framework of the forecast. However, The author mistakenly treated "false alarms" as when an event is not observed at a grid point and forecast within a neighborhood framework of the forecast. We sincerely apologize for this error, which has now been corrected throughout the entire revised manuscript. Fig. R5 presents the BIAS scores for the case study mentioned in the original manuscript, and the values have become reasonable.

[Figure]

**Figure R5** The bias score (BIAS; (a)–(c)) of the predicted hourly accumulated precipitation of the CTRL (the blue solid line) and DA-W (the red solid line) experiments for thresholds of 1 mm h⁻¹, 5 mm h⁻¹, and 20 mm h⁻¹ for the heavy rainfall case on 4 July 2020.

Fig. 4: The flowchart is a little bit confusing to me. Is it continuous cycling or partial cycling? Were the assimilations done only at 06Z every day? The description of the flowchart is not clear enough to me.

**Response:** We apologize for this confusing. It is not a continuous cycling, as assimilation is only performed at 0600 UTC every day. However, we have revised the batch experiments, and the corresponding flowchart has also been updated as Fig. 3 (b) in the revised manuscript.

Fig. 5 and 6: The authors may enhance clarity by presenting the average forecast skills of ETS, FSS, and BIAS over a ten-day cycling period instead of displaying

results for each case. By utilizing ten days' samples, the inclusion of error bars in the figures can provide a more comprehensive representation of variability and uncertainty.

**Response:** Thank you for your suggestion. We have rerun the batch experiments, and the revised manuscript includes the 10-day (July 1 to July 10, 2020) averaged ETS, FSS, and BIAS scores as suggested by the reviewer (Figs. 4-6). Additionally, error bars have been included in the figures to provide a more comprehensive representation of variability and uncertainty.

We thank the reviewer again for taking the time to review our manuscript.

**References**

Clark, A. J., Gallus, W. A. Jr., Weisman, M. L.: Neighborhood-based verifcation of precipitation forecasts from convection-allowing NCAR WRF Model simulations and the operational NAM, Wea. Forecasting, 25:1495–1509, doi: 2010WAF2222404.1, 2010.

Chen, Z., Sun, J., Qie, X., Zhang, Y., Ying, Z., Xiao, X., and Cao, D.: A method to update model kinematic states by assimilating satellite-observed total lightning data to improve convective analysis and forecasting, J. Geophys. Res. Atmos., 125(22), 1–26, doi:10.1029/2020jd033330, 2020.

Xiao, X., Sun, J., Qie, X., Ying, Z., Ji, L., Chen, M., and Zhang, L.: Lightning data assimilation scheme in a 4DVAR system and its impact on very short-term convective forecasting, Mon. Weather Rev., 149(2), 353–373, doi:10.1175/mwr-d-19-0396.1, 2021.

**Response to Reviewer #2**

Reviewer #2

The article presents a 3DVAR assimilation scheme for w, which appears reasonable and has a positive impact based on results from a heavy-rain event and a 10-day batch experiment.

The authors sincerely appreciate the valuable time invested by the reviewer in reviewing our manuscript. We have addressed your queries in the subsequent responses.

However, there are some doubts:

1. From the batch experiment results, it was found that w assimilation has a small impact on the forecast. However, in the case study where w was assimilated in a cyclical manner, it led to significant improvements in the forecast. It is suggested that the authors also set multiple w assimilations in the batch experiment section.

**Response**: Thank you very much for your comments and professional advice. Based on your suggestion, we have conducted a new set of batch experiments to explore multiple assimilation iterations for vertical velocity ($w$). In Figure 3 (b) of the revised manuscript, both the CTRL and DA-W experiments were initialized at 0000 UTC daily from July 1 to July 10, 2020, and run until 1200 UTC each day. The first 3 hours were considered as a "spin-up" period. In the CTRL experiments, observations from aircraft measurements, radiosondes, and other sources (for a comprehensive list, refer to Fig. 3 (b)) were assimilated from 0300 to 0600 UTC with a 1-hour assimilation interval (radial velocity observations are available at each analysis time, while other data sources are only available at 0300 and 0600 UTC). The CTRL-1CY experiment indicates assimilation at 0300 UTC only, while the CTRL-2CY experiment represents assimilation at 0300 and 0400 UTC, and so on (the number preceding the experiment name "CY" represents the assimilation iterations). The DA-W experiments are similar to the CTRL experiments, but include the assimilation of $w$ pseudo-observations ($w$ pseudo-observations are available at each analysis time).

[Figure]

**Figure 3 (b) of the revised manuscript.** Illustration of the numerical experimental scheme for the CTRL and DA-W batch experiments. Both experiments utilize NCEP GFS data as the initial condition (IC) and boundary condition (BC). The abbreviation "Fcst" represents forecast. The assimilated data comprises conventional observations from aircraft measurements (AIREP), radiosondes (TEMP), ships (SHIP), and ground stations (SYNOP). In addition, cloud-track-wind (SATOB), precipitable water derived from the Global Positioning System (GPSPW), refractivity radio-occultation data from the Global Navigation Satellite System (GNSS-RO), wind profiler radar (WPR), velocity-azimuth display (VAD) wind, and the radar radial velocity (VR) are assimilated. The pseudo-$w$ data is also assimilated for the DA-W experiments.

Figs. 4-6 in the revised manuscript present the 10-day averaged forecast skills for hourly accumulated precipitation from 0600 UTC to 1200 UTC. For the threshold of 1 mm h$^{-1}$, it is not always the case that the ETS score improves as the number of assimilation times increases for both CTRL and DA-W experiments. However, with an increase in the scoring threshold, especially for 20 mm h$^{-1}$, a higher score is generally achieved with more assimilation times, indicating a positive impact of multiple assimilation on the forecast. However, with an increase in the scoring threshold, especially for 20 mm h$^{-1}$, a higher score is generally achieved with more assimilation times, indicating a positive impact of multiple assimilation on the forecast. When comparing the ETS scores of the CTRL and DA-W experiments with the same assimilation times, it can be seen that the DA-W experiment has a neutral or slightly worse effect on the forecast at threshold of 1 mm h$^{-1}$ compared to the CTRL experiment. However, at thresholds of 5 and 20 mm h$^{-1}$, the DA-W experiment achieves higher scores than the CTRL experiment in most situations, regardless of multiple or single assimilation. Moreover, the experiment with 3 assimilation times (denoted by experiment names ending with "3CY") demonstrate the most significant

improvements compared to experiments with other assimilation times.

The FSS scores provide clearer results: for experiments with the same assimilation times in CTRL and DA-W, the DA-W experiment consistently achieves better scores, indicating that the assimilation of $w$ has a better adjustment effect on the forecast of precipitation location. From the BIAS scores, the DA-W experiments have a neutral impact on the forecast compared to the CTRL experiments. In the first 3-hour forecast, the DA-W experiment generally performs worse than the CTRL experiment (with the same assimilation times) for each threshold value, primarily due to producing more false alarms. However, in the latter 3-hour forecast, the DA-W experiment demonstrates better scores compared to the CTRL experiment.

[Figure]

**Figure 4 of the revised manuscript.** The 10-d (July 1 to July 10, 2020) averaged equitable threat score (ETS; solid dots) of the predicted hourly accumulated precipitation from 0600-1200 UTC of the CTRL and DA-W experiments for thresholds of (a) 1 mm h$^{-1}$, (b) 5 mm h$^{-1}$, and (c) 20 mm h$^{-1}$. The top (bottom) of the line that passes through the solid dot corresponds to the maximum (minimum) ETS value for those 10 days.

[Figure]

**Figure 5 of the revised manuscript.** Same as Fig. 4 but for the neighborhood-based fractions skill score (FSS).

[Figure]

**Figure 6 of the revised manuscript.** Same as Fig. 4 but for the bias score (BIAS). The black dashed line represents BIAS value equals 1.

2. As mentioned by the author, some studies assimilate w from total lightning data. In this article, however, the w observations are derived from radar reflectivity data. It is unclear how the authors obtained at the approximate magnitude of the w values using this method.

**Response**: Yes, in this study, the pseudo-$w$ observations are derived from radar reflectivity data. The authors conducted statistical analysis of the pseudo-$w$ observations. From the frequency distribution of the pseudo-$w$ observations in different value and height bins (Fig. R6 (a)–(d)), it is observed that the majority (90.4%) of pseudo-$w$ observations have small values, ranging from 0.0 to 0.6 m s$^{-1}$.

[Figure]

**Figure R6** The frequency distribution of pseudo-$w$ observations at (a) 1200 UTC, (b) 1300 UTC, (c) 1400 UTC, and (d) 1500 UTC on July 4, 2020.

In addition, were the authors able to compare the radar reflectivity-derived w with the w of the model background field to determine any differences in magnitude between the two values? If there is a significant difference, it may be necessary to remove the larger w values during the assimilation process.

**Response**: We appreciate the valuable suggestion from the reviewer. Fig. R7 presents the mean $w$ profiles (blue solid lines) calculated from the background at different analysis times for the case studied in our manuscript (please refer to the caption of Fig. R7 for specific area boundaries. The statistics are calculated based on grid points where the composite radar reflectivity exceeds 35 dBZ). The shape of the profile shows good agreement with the results of Yuter and Houze (1995). In the convection area, the mean $w$ increases from low values at lower levels to a peak value occurring at middle to upper levels (approximately 6-8 km), and then decreases at higher altitudes. The maximum mean $w$ of the background is about 0.5 m s$^{-1}$ at most analysis times, except for 1500 UTC, which is almost 1.0 m s$^{-1}$. These values are close in magnitude to the pseudo-$w$ observation values, of which 90.4% are smaller than 0.6 m s$^{-1}$.

[Figure]

**Fig. R7** The distribution of pseudo-*w* observations (red dots) retrieved from radar reflectivity data at different analysis times on July 4, 2020. The blue solid lines represent the averaged *w* profiles calculated from the background field (xb), and the averaging is performed over the area of (35–44°N, 113–119°E).

Some minor revisions are as follows:
Page 1:
Line 21: Change "the result indicates" to "the results indicate".
**Response**: Thank you for your suggestion. This sentence has been removed in the revised manuscript.

Line 22: The statement "leading to improved equitable threat score (frequency skill score) for the first 1 h (3 h) precipitation forecasts" may cause confusion, please describe it in detail.
**Response**: Thank you for your suggestion. Since we reran the experiments, this sentence has been removed in the revised manuscript.

Line 23: Change "assimilated" to "assimilation".
**Response**: Thank you for your suggestion. This sentence has been removed in the revised manuscript.

Page 2:
Line 36: Change "allows they" to "allows them".
**Response**: Thank you for your suggestion. Revised as suggested (Line 40 in the revised manuscript).

Line 41-42: Delete "of vertical velocity".
**Response**: Thank you for your suggestion. Deleted as suggested (Line 45 in the revised manuscript).

Line 48: Add "s" to the word "field".
**Response**: Thank you for your suggestion. It has been revised as suggested (Line 52 in the revised manuscript).

Page 3:
Line 75: Delete "real".
**Response**: Thank you for your suggestion. This sentence has been removed in the

revised manuscript.

Lines 89-91: This sentence is quite difficult to understand. I suggest that it be described simply and clearly.
**Response**: We apologize for any confusion caused. We have revised the related text to "The observation operator H is used to derive the equivalent of the observations from model state variables" in order to improve clarity. (Line 94 in the revised manuscript).

Line 92: Delete "to assimilate w observation directly".
**Response**: Thank you for your suggestion. Deleted as suggested (Line 95 in the revised manuscript).

Line 96: Add "s" to the word "adjust".
**Response**: Thank you for your suggestion. The word "adjust" has been revised to "enables" (Line 99 in the revised manuscript).

Page 14:
Line 279: Change "wish" to "wishes".
**Response**: Thank you for your suggestion. It has been revised as suggested (Line 315 in the revised manuscript).

We would like to thank the reviewer for taking the time to review our manuscript.

**References**

Yuter, S. E. and Houze, R. A.: Three-dimensional kinematic and microphysical evolution of Florida cumulonimbus: Part II. Frequency distribution of vertical velocity, reflectivity, and the differential reflectivity, Mon. Weather Rev., 123, 1941–1963, doi:10.1175/1520-0493(1995)123<1941:TDKAME>2.0.CO;2, 1995.

---

## Referee Report (RR1)

This is my second review of this manuscript. The authors re-run the batch experiments with increased assimilation times, and the experimental design became more reasonable. I appreciate the authors' modifications to the manuscript. I recommend publication with some minor changes and residual explanations, which I have highlighted below.

**Scientific and major points**

1、 The author's response indicates that the pseudo-vertical velocity observation values used for assimilation are generally small, usually less than 1 m/s, which closely matches the magnitude of the background field. Additionally, the frequency distribution of these observations suggests that numerous pseudo-vertical velocities are assimilated for each analysis time, numbering in the thousands. The case study (Figure 8 of the revised manuscript) demonstrates the positive impact of these observations on precipitation adjustments, could you please provide me with the value of the horizontal wind analysis increments resulting from assimilating such a large number of observations?

2、 The authors emphasize the propensity for generating some false precipitation forecasts when pseudo-vertical velocity observations are assimilated, as evidenced by the outcomes of batch experiments. Is this caused by the use of a large horizontal influence radius?

Line 215: "Forecasts with higher ETS (close to 1) and FSS (close to 1) and lower BIAS (closer to 1), demonstrate better forecast skills." The statement about the BIAS score is not rigorous. A lower BIAS score does not indicate better forecast skills.

**Minor points**

Line 90: Add "field" after "background".

Line 151: Add "component" after "horizontal wind".

Line 152: From Figure 2, the convergence of u wind is not extending to ground, but 1000 hPa?

Line 174: Change "vertical velocities" to "pseudo-$w$ observations".

Lines 176-179: Add the definition of variables $Z$ and $H$. In addition, replace the character $H$ with another symbol to distinguish it from the observation operator symbol in the manuscript.

Line 209: Add "observations" after "(VR)".

Line 212: In this section, the assessment for batch experiments is not limited to convective precipitation. It is recommended to delete the word "convective".

Line 235: Change "0600-1200 UTC" to "0600 to 1200 UTC".

Line 281: "while a horizontal wind divergence".

Line 282: Delete "effectively".

Line 285: Change "using" to "based on".

Line 288: "assimilation (DA-W) experiments" such a statement may lead to ambiguity. In fact, the control experiment in the article also involves observation assimilation. Here, it would be better to highlight that the DA-W experiment assimilates vertical velocity.

---

## Referee Report (RR2)

The authors have addressed most comments, and the quality of the manuscript improves a lot. I suggest publishing it after a minor revision.

**Minor revision:**
Line 66: "assimilating w observations as a control variable" is not accurate, replace it with "assimilating w observations with control variable w".

Lines 70-71: "This operator ensures … the 3DVar cost function". This sentence is a little bit confusing to me. The observation operator links the model state variables (not control variable) to the observed variables. How about "This operator ensures adherence to physical constraints and links the w observations to other model state variables for the minimizing the 3D-Var cost function".

Lines 86-92: The Cv, d, R, x, xb and B are either vectors or matrixs, so they should be bold and non-italic. Please check it throughout the manuscript.

Figure 2: As I mentioned in previous review, adding a constraint to the vertical propagation of w assimilation would have an impact. It would be better if some discussions are given here even though it is not implemented.

Lines 184-185: It should be "a series of continuous 10-day runs … were".

Line 215: it is a little bit confusing "lower BIAS" here, better to be "closer BIAS to 1".

Lines 221-222: delete "compared to the CTRL experiment".

Line 228: better to modify "has a better adjustment effect" to "has a positive impact" to be clearer.

Lines 260-264: Could you discuss more on it. A deeper discussion could benefit more to this paper.

---

## Author Response (AR2)

**Response to reviewers of the manuscript**

"*A 3D-Var Assimilation Scheme for Vertical Velocity with the CMA-MESO v5.0*"

H. Li, Y. Yang, J. Sun, Y. Jiang, R. Gan, and Q. Xie

for Geoscientific Model Development

**Response to Reviewer #1**

Reviewer #1

This is my second review of this manuscript. The authors re-run the batch experiments with increased assimilation times, and the experimental design became more reasonable. I appreciate the authors' modifications to the manuscript. I recommend publication with some minor changes and residual explanations, which I have highlighted below.

**Response:** We gratefully thank the reviewer for your valuable feedback that we have used to improve the quality of our manuscript. Our response and changes/additions to the manuscript are given in the blue text below.

**Scientific and major points**:

1、The author's response indicates that the pseudo-vertical velocity observation values used for assimilation are generally small, usually less than 1 m/s, which closely matches the magnitude of the background field. Additionally, the frequency distribution of these observations suggests that numerous pseudo-vertical velocities are assimilated for each analysis time, numbering in the thousands. The case study (Figure 8 of the revised manuscript) demonstrates the positive impact of these observations on precipitation adjustments, could you please provide me with the value of the horizontal wind analysis increments resulting from assimilating such a large number of observations?

**Response:** Thank you for your valuable comments. Figure R1 presents the horizontal wind increments for the case study in our manuscript at the analysis time 0600 UTC on July 9, 2020. At the lower level of the model (Fig. R1 (a)), there are obvious horizontal wind (up to a maximum of about 10 m $s^{-1}$) and convergence (less than $-4 \cdot 10^{-4}$ $s^{-1}$) increments. At the same time, there are divergence or weak wind convergence increments in the middle level of the model (Fig. R1 (b)), and such a configuration of the horizontal wind field facilitates the model to produce certain vertical velocities in the middle and lower levels, thus further facilitating convection at these locations.

[Figure]

**Figure R1.** The analysis increments of horizontal wind (vector; unit: m s$^{-1}$) and horizontal wind divergence (color; unit: $10^{-4}$ s$^{-1}$) at the (a) 13th (~850 hPa) and (b) 23th (~500 hPa) levels of the model at 0600 UTC on July 9, 2020.

2、 The authors emphasize the propensity for generating some false precipitation forecasts when pseudo-vertical velocity observations are assimilated, as evidenced by the outcomes of batch experiments. Is this caused by the use of a large horizontal influence radius?

**Response:** Yes, we agree. In our study, a relatively large scale factor of 400 km was used for several reasons. First, the derived $w$ pseudo-observations are mostly small values, i.e., smaller than 1 m s$^{-1}$. Second, the observation operator for $w$, the adiabatic Richardson equation, combines the continuity equation, adiabatic equations, and the hydrostatic relation, making it more suitable for large-scale systems.

Line 215: "Forecasts with higher ETS (close to 1) and FSS (close to 1) and lower BIAS (closer to 1), demonstrate better forecast skills." The statement about the BIAS score is not rigorous. A lower BIAS score does not indicate better forecast skills.

**Response:** Thank you for your valuable comments, and we apologize for our less rigorous statement. The statement "Forecasts with higher ETS (close to 1) and FSS (close to 1) and lower BIAS (closer to 1), demonstrate better forecast skills." has been revised to "Forecasts with higher ETS (close to 1) and FSS (close to 1) and closer BIAS to 1, demonstrate better forecast skills." (Line 227 in the revised manuscript).

**Minor points:**

Line 90: Add "field" after "background".

**Respon**se: Thank you for your suggestion. It has been revised as suggested (Line 90 in the revised manuscript).

Line 151: Add "component" after "horizontal wind".

**Respon**se: Thank you for your suggestion. It has been added as suggested (Line 158 in the revised manuscript).

Line 152: From Figure 2, the convergence of u wind is not extending to ground, but

1000 hPa?

**Response:** Yes, our previous statement was not precise enough. According to the reviewer's suggestion, we have revised the sentence to "there is a convergence of $u$ wind that extends to the 1000 hPa" (Line 159 in the revised manuscript).

Line 174: Change "vertical velocities" to "pseudo-$w$ observations".

**Response:** Thank you for your suggestion. It has been revised as suggested (Line 185 in the revised manuscript).

Lines 176-179: Add the definition of variables $Z$ and $H$. In addition, replace the character $H$ with another symbol to distinguish it from the observation operator symbol in the manuscript.

**Response:** Thank you for your suggestion. The definition of symbol $Z$ has been added in the revised manuscript (Line 189 in the revised manuscript). Following the reviewer's suggestion, we have revised the symbol $H$ to $H_{ei}$, and the definition of symbol $H_{ei}$ has been added in the revised manuscript (Line 189 in the revised manuscript).

Line 209: Add "observations" after "(VR)".

**Response:** Thank you for your suggestion. It has been added as suggested (Line 221 in the revised manuscript).

Line 212: In this section, the assessment for batch experiments is not limited to convective precipitation. It is recommended to delete the word "convective".

**Response**: Thank you for your suggestion. Deleted as suggested (Line 224 in the revised manuscript).

Line 235: Change "0600-1200 UTC" to "0600 to 1200 UTC".

**Response:** Thank you for your suggestion. It has been revised as suggested (Line 247 in the revised manuscript).

Line 281: "while a horizontal wind divergence".

**Response:** Thank you for your suggestion. It has been revised as suggested (Line 301 in the revised manuscript).

Line 282: Delete "effectively".

**Response**: Thank you for your suggestion. Deleted as suggested (Line 302 in the revised manuscript).

Line 285: Change "using" to "based on".

**Response:** Thank you for your suggestion. We have revised it as the reviewer suggested (Line 304 in the revised manuscript).

Line 288: "assimilation (DA-W) experiments" such a statement may lead to ambiguity. In fact, the control experiment in the article also involves observation assimilation. Here, it would be better to highlight that the DA-W experiment assimilates vertical velocity.

**Response:** Thank you for your suggestion. Following the reviewer's suggestion, this unrigorous description has been revised to "Two sets of experiments were configured, including CTRL and DA-W experiments with different assimilation iterations. Both sets of experiments assimilated aircraft measurements, radiosondes, and other observations (for a comprehensive list, refer to Fig. 3 (b)) at 1-hour intervals during a 3-hour data assimilation period. In addition, the pseudo-$w$ observations are also assimilated in the DA-W experiments." (Lines 306–309 in the revised manuscript) to make it clearer.

Thanks again for the reviewer's time and valuable comments.

**Response to Reviewer #2**

Reviewer #2

The authors have addressed most comments, and the quality of the manuscript improves a lot. I suggest publishing it after a minor revision.

The authors sincerely appreciate the valuable time the reviewer has dedicated to reviewing our manuscript. Below, we have outlined the detailed corrections made to our previous draft.

**Minor revision**:

Line 66: "assimilating w observations as a control variable" is not accurate, replace it with "assimilating w observations with control variable w".

**Response**: Thank you for your suggestion. We have revised it as you suggested (Line 65 in the revised manuscript).

Lines 70-71: "This operator ensures … the 3DVar cost function". This sentence is a little bit confusing to me. The observation operator links the model state variables (not control variable) to the observed variables. How about "This operator ensures adherence to physical constraints and links the w observations to other model state variables for the minimizing the 3D-Var cost function".

**Response**: Thank you for your suggestion, and we apologize for our less rigorous statement. We have revised the text as you suggested. Please see Lines 69–70 in the revised manuscript.

Lines 86-92: The Cv, d, R, x, xb and B are either vectors or matrixs, so they should be bold and non-italic. Please check it throughout the manuscript.

**Response**: Thank you for your careful consideration. Following the reviewer's suggestion, we have updated the symbol Cv, d, R, x, xb and B to be bold and non-italic throughout the revised manuscript.

Figure 2: As I mentioned in previous review, adding a constraint to the vertical propagation of w assimilation would have an impact. It would be better if some discussions are given here even though it is not implemented.

**Response**: Yes, we think this is a good suggestion. We have added the statement "It is worth noting that there are currently no constraints on the $w$ assimilation impact propagation in the vertical direction. However, it is better to set limits to prevent excessive increments at higher model levels, thus leading to more realistic forecasts." in the revised manuscript (Lines 160–162) as you suggested.

Lines 184-185: It should be "a series of continuous 10-day runs … were".

**Response**: Thank you for your suggestion. It has been revised as suggested (Lines 196–197 in the revised manuscript).

Line 215: it is a little bit confusing "lower BIAS" here, better to be "closer BIAS to

1".

**Response**: Thank you for your suggestion. It has been revised as suggested (Line 227 in the revised manuscript).

Lines 221-222: delete "compared to the CTRL experiment".

**Response**: Thank you for your suggestion. Deleted as suggested (Line 233 in the revised manuscript).

Line 228: better to modify "has a better adjustment effect" to "has a positive impact" to be clearer.

**Response**: Thank you for your suggestion. It has been revised as suggested (Line 240 in the revised manuscript).

Lines 260-264: Could you discuss more on it. A deeper discussion could benefit more to this paper.

**Response**: Thank you for your suggestion. The statement has been revised to "In Fig. 8(a), line A-B represents the observed main precipitation belt. Fig. 9 shows the sections along the line A-B for the CTRL-4CY and DA-W-4CY experiments at 0700 UTC on July 9, 2020. The DA-W-4CY experiment effectively enhances the $w$ values across the entire model layers. This enhancement is achieved by generating increments of wind convergence (less than $-4 \times 10^{-4}$ s$^{-1}$) at the lower (the 13th) level of the model, while inducing divergence or weak wind convergence increments at the middle (the 23th) level of the model (Fig. 10). Such a configuration of the horizontal wind field enables the model to generate specific vertical velocities in the middle and lower levels, leading to a decrease in water vapor below 850 hPa compared to the CTRL-4CY experiment (Fig. 9(c)). Simultaneously, positive increments of water vapor are observed in the middle and upper layers of the model. Consequently, upward movement enhances the vertical transport of water vapor, promoting water vapor saturation and facilitating cloud formation, ultimately resulting in rainfall." (Lines 272–280 in the revised manuscript) to have a deeper discussion on it.

We would like to thank you again for taking the time to review our manuscript.

[Figure]

**Figure 10 of the revised manuscript.** The analysis increments of horizontal wind (vector; unit: m s$^{-1}$) and horizontal wind divergence (color; unit: $10^{-4}$ s$^{-1}$) of the (a, b) CTRL-4CY and (c, d) DA-W-4CY experiments at the (a, c) 13th (~850 hPa) and the (b, d) 23th (~500 hPa) model levels at 0600 UTC on July 9, 2020.

**Response to Reviewer #3**

Reviewer #3

In this study, a 3D-Var data assimilation scheme for vertical velocity, based on the adiabatic Richardson equation is developed within the high-resolution CMA-MESO model, enabling the update of horizontal winds and mass fields of the model's background. The manuscript is well written, the experiments are well design, and the conclusions are well supported by the results. However, some details should be clarified and the manuscript also should be further improved according to the comments given below. I think it is suitable for publication after moderate to major revisions.

We sincerely appreciate the time and effort invested by you in evaluating our manuscript. We have carefully considered your valuable suggestions and have made extensive revisions to our previous draft to address the issues you raised. The detailed corrections are outlined below for your review. However, before receiving your review comments, the manuscript had already undergone one round of revisions. Some of these modifications may address your concerns. Hence, in subsequent responses, we will refer to the initial revised manuscript as "the revised manuscript v1". We appreciate your thorough feedback and believe that these revisions have significantly improved the quality of our work. Thank you once again for your insightful suggestions.

**Major comments**:

1. The main verification method in this study is precipitation score, attention also needs to be paid to how precipitation is distributed at one or two typical moments.

**Response:** Thank you for your valuable comments. Based on the previous feedback from other reviewers, we have conducted a new set of batch experiments. Both the CTRL and DA-W experiments were initialized at 0000 UTC daily from July 1 to July 10, 2020, and run until 1200 UTC each day. The first 3 hours were considered as a "spin-up" period. In the CTRL experiments, observations from aircraft measurements, radiosondes, and other sources (for a comprehensive list, refer to Fig. 3 (b)) were assimilated from 0300 to 0600 UTC with a 1-hour assimilation interval (radial velocity observations are available at each analysis time, while other data sources are only available at 0300 and 0600 UTC). The CTRL-1CY experiment indicates assimilation at 0300 UTC only, while the CTRL-2CY experiment represents assimilation at 0300 and 0400 UTC, and so on (the number preceding the experiment name "CY" represents the assimilation iterations). The DA-W experiments are similar to the CTRL experiments, but include the assimilation of $w$ pseudo-observations ($w$ pseudo-observations are available at each analysis time).

[Figure]

**Figure 3 (b) of the revised manuscript v1.** Illustration of the numerical experimental scheme for the CTRL and DA-W batch experiments. Both experiments utilize NCEP GFS data as the initial condition (IC) and boundary condition (BC). The abbreviation "Fcst" represents forecast. The assimilated data comprises conventional observations from aircraft measurements (AIREP), radiosondes (TEMP), ships (SHIP), and ground stations (SYNOP). In addition, cloud-track-wind (SATOB), precipitable water derived from the Global Positioning System (GPSPW), refractivity radio-occultation data from the Global Navigation Satellite System (GNSS-RO), wind profiler radar (WPR), velocity-azimuth display (VAD) wind, and the radar radial velocity (VR) are assimilated. The pseudo-$w$ data is also assimilated for the DA-W experiments.

In addition, following the previous reviewer's suggestion, we have replaced the individual test with a case in the batch experiments, specifically the case initialized on July 9, 2020. For this case, we have also included the distribution of its 6-hour accumulated precipitation as Fig. 8 in the revised manuscript v1.

[Figure]

**Figure 8 of the revised manuscript v1.** The 6-hour (0600-1200 UTC) accumulated precipitation (units: mm) on July 9, 2020 for (a) observations (OBS), (b) CTRL-4CY, and (c) DA-W-4CY experiments. The areas enclosed by dotted purple lines indicate regions with observed strong rainfall.

2. Also, how does the assimilated vertical velocity affect the horizontal wind fields and how does it affect other thermodynamic fields such as temperature and humidity?

It is suggested to add one or two figures of quantitative and qualitative verification about the thermodynamic fields.

**Response:** Thank you for your suggestion. In this study, the adiabatic Richardson equation, which links $w$ observations to model state variables $u$, $v$, and $\Pi$ ( the dimensionless pressure), is used as the observation operator for $w$. Therefore, assimilating $w$ observations directly generates analysis increments for $u$, $v$, and $\Pi$ variables, and then the analysis increments of temperature and specific humidity are calculated from the analysis increment of $\Pi$ based on the physical constraint equations (state equation of moist air and hydrostatic balance equation). A single $w$ observation experiment is conducted in our manuscript as Fig. 1, the pseudo-observation of $w$ is positioned at an altitude of 5448.6 m (23th model level, approximately 500 hPa) with a value of 1 m s$^{-1}$. In order to quantify the size of the increments, the following description "The increments of horizontal wind at the lower and middle model levels can reach 0.060 and 0.077 m s$^{-1}$, respectively." (Lines 151–152 in the revised manuscript) has been added in the revised manuscript. In addition, the statements "($\sim$ -8.7×10$^{-6}$–2.6 ×10$^{-5}$ K in Fig. 1 (d))" and "($\sim$ -7.0×10$^{-8}$–6.4 ×10$^{-8}$ kg kg-1 in Fig. 1 (c))" (Lines 155–156 in the revised manuscript) to quantify the increments of temperature and humidity fields.

3. In particular, how the assimilation of affects vertical velocity fields itself, related verifications are also suggested.

**Response:** Thank you very much for your comments and professional advice. Since $w$ is not an analysis variable, it is not updated at the analysis step. In our study, the adiabatic Richardson equation is used as the observation operator to assimilate pseudo-$w$ observations, which leading to the update of the horizontal winds and pressure fields as the $w$ assimilated. Although $w$ is not directly updated at the analysis time, due to the assimilation of vertical information updating the horizontal wind field, $w$ will be adjusted during subsequent model integrations through constraints imposed by the dynamic fields.

4. Since this study mainly focused on vertical velocity, vertical distribution plot of analysis increments in single observation test is suggested.

**Response:** Thank you for your suggestion. The vertical cross-sections of the analysis increment have been included as Fig. 2 in the revised manuscript v1. Additionally, the following statement has been added in the revised manuscript v1 (Lines 156–160): "From the vertical cross-section of the analysis increment for each variable (Fig. 2), it can be seen that the increase in specific humidity is primarily concentrated in the lower layer below the observation location, while the increases in the other three variables are distributed throughout the entire layer. Regarding the increase in horizontal wind component $u$, below the single point observation, there is a convergence of $u$ wind that extends to the 1000 hPa. Above the single point observation, there is a divergence of $u$ wind that extends to approximately 150 hPa".

[Figure]

**Figure 2 of the revised manuscript v1**. The analysis increments of (a) zonal wind $u$, (b) meridional wind $v$ (unit: m s$^{-1}$), (c) specific humidity (unit: kg kg$^{-1}$), and (d) temperature (unit: K) in a vertical cross-section at 38.0° N at 1500 UTC on July 4, 2020, for the single observation experiment. The solid black dots in the figure represent the locations of the single observation.

5. The treatment of the observation operator in this study is more like a static initialization based on equilibrium equations. If the model also assimilates the horizontal wind field at the same time, how should the analytical field of the final wind field take advantage of the different information. Does the difference between the two information create spurious gradients?

**Response:** We appreciate your valuable comment. To illustrate this issue, a comparison of the horizontal wind increments for two experiments (experiments CTRL-4CY and DA-W-4CY) in our revised manuscript v1 is presented as Figure R2. Please refer to the response to question 1 for a detailed description of experiments CTRL-4CY and DA-W-4CY. The CTRL-4CY experiment involves assimilation of horizontal wind data from other sources (radiosonde data) as mentioned by the reviewer. The DA-W-4CY experiment, on the other hand, assimilates vertical velocity in addition to the CTRL-4CY experiment, thereby adjusting the model's horizontal wind fields. From Figure R2, the horizontal wind analysis increments produced by the CTRL-4CY experiment are comparable to those of the DA-W-4CY experiment, but at locations with pseudo-$w$ observations (as shown in Fig. R3), the DA-W-4CY experiment exhibits larger horizontal wind increments, accompanied by noticeable convergence and divergence patterns. At the same time, at locations without

pseudo-*w* observations, the horizontal wind analysis increments of the DA-W-4CY experiment are comparable to those of the CTRL-4CY experiment.

[Figure]

**Figure R2.** The analysis increments of horizontal wind (vector; unit: m s⁻¹) and horizontal wind divergence (color; unit: $10^{-4}$ s⁻¹) of the (a, a1) CTRL-4CY and (b, b1) DA-W-4CY experiments at the (a, b) 13th (~850 hPa) and the (a1, b1) 23th (~500 hPa) model levels at 0600 UTC on July 9, 2020. (c) and (c1) are the difference between the CTRL-4CY and DA-W-4CY experiments.

[Figure]

**Figure R3.** The derived maximum *w* pseudo-observations (units: m s⁻¹) at 0600 UTC on July 9, 2020.

Figure R4 illustrates the evolution of gradients during the minimization process for the CTRL-4CY and DA-W-4CY experiments. It can be seen that the DA-W-4CY experiment generates larger gradients than the CTRL-4CY experiment at the initial iteration step, but it rapidly converges, and even completes the minimization process more quickly than the CTRL-4CY experiment.

[Figure]

**Figure R4.** The evolution of gradient values with the iteration steps at the analysis time 0600 UTC July 9, 2020 for the CTRL-4CY and DA-W-4CY experiments.

**Minor comments**:

1. This study uses radar reflectivity-derived vertical velocities as observations, in my opinion, radar radial wind also contains vertical velocity information. Could the authors add some discussions regarding the future use of radar radial winds for vertical velocity assimilation?

**Response:** Thank you for your valuable comments. Based on your suggestion, we have added "In addition, the radial velocity also includes vertical velocity information." in the revised manuscript (Line 322).

2. On page 3, section 2.1, the description of the CMA-MESO 3D-Var system should be more specific.

**Response:** Thank you for your suggestion. The description of the CMA-MESO 3D-Var system has been revised in our revised manuscript v1: 1) in order to ensure consistency with Eq. (10), we have modified the cost function $J$ to take the form of control variable; 2) the statement "the best analysis $\mathbf{x}$ can be derived from the control variable $\mathbf{c_v}$ (the control variables for CMA-MESO include the zonal and meridional winds, pseudo-relative humidity, temperature, and surface pressure) by minimizing a cost function $J$ of $\mathbf{c_v}$" has been added (Lines 85–89 in the revised manuscript).

3. On page 4, line 106, the insertion of the parameter expression K=cp/R is too abrupt. Please add some description before and after as appropriate.

**Response:** Thank you for your suggestion. It has been revised to "The parameter $\kappa$ in Eq. (5) can be expressed as $\kappa = {}^{c_p}/_R$." to make it more appropriate (Line 108 in the revised manuscript).

4. The legend for CTRL and DA-W in the upper left corner of Figure 3(a) should be aligned.

**Response:** Thank you for your suggestion. This figure has been removed in the revised manuscript v1.

5.  Figures 4 (a) and 4 (b) can be presented in two figures.

**Response:** We appreciate your careful attention. Nevertheless, since our revised manuscript already contains 10 figures, we propose merging these two figures (Figures 3 (a) and 3 (b)) into one, taking into account the overall figure count.

6.  Figure 4(b) does not show enough differences for the CTRL vs. DA-W group.

**Response:** Thank you for your suggestion. Based on the previous feedback from other reviewers, we have conducted a new set of batch experiments. Figure 4(b) now is Fig. 3 (b) in the revised manuscript v1. Both the CTRL and DA-W experiments were initialized at 0000 UTC daily from July 1 to July 10, 2020, and run until 1200 UTC each day. The first 3 hours were considered as a "spin-up" period. In the CTRL experiments, observations from aircraft measurements, radiosondes, and other sources (for a comprehensive list, refer to Fig. 3 (b)) were assimilated from 0300 to 0600 UTC with a 1-hour assimilation interval (radial velocity observations are available at each analysis time, while other data sources are only available at 0300 and 0600 UTC). The CTRL-1CY experiment indicates assimilation at 0300 UTC only, while the CTRL-2CY experiment represents assimilation at 0300 and 0400 UTC, and so on (the number preceding the experiment name "CY" represents the assimilation iterations). The DA-W experiments are similar to the CTRL experiments, but include the assimilation of $w$ pseudo-observations ($w$ pseudo-observations are available at each analysis time).

[Figure]

**Figure 3 (b) of the revised manuscript v1.** Illustration of the numerical experimental scheme for the CTRL and DA-W batch experiments. Both experiments utilize NCEP

GFS data as the initial condition (IC) and boundary condition (BC). The abbreviation "Fcst" represents forecast. The assimilated data comprises conventional observations from aircraft measurements (AIREP), radiosondes (TEMP), ships (SHIP), and ground stations (SYNOP). In addition, cloud-track-wind (SATOB), precipitable water derived from the Global Positioning System (GPSPW), refractivity radio-occultation data from the Global Navigation Satellite System (GNSS-RO), wind profiler radar (WPR), velocity-azimuth display (VAD) wind, and the radar radial velocity (VR) are assimilated. The pseudo-$w$ data is also assimilated for the DA-W experiments.

7. On page 4, section 2.3 Accuracy check, the presentation of the judgment logic of the adjoint test under double precision could be made clearer, as is your account of verification of gradient correctness in the second half of the section.

**Response:** Thank you for your suggestion. According to your comments, the authors have made revisions to these two sections for clarity, as shown below (Lines 123–141 in the revised manuscript):

After completion of the $w$ observation operator, the correctness of the adjoint operator should be checked (adjoint check). For the tangent linear $\mathbf{H}$ and its adjoint $\mathbf{H}^{\mathrm{T}}$ of an observation operator, the following formula is always satisfied:

$$< \mathbf{H}(\delta x), \mathbf{H}(\delta x) >=< \mathbf{H}^{\mathrm{T}}\big(\mathbf{H}(\delta x)\big), \delta x >, \tag{9}$$

where $\delta x$ represents a small perturbation and $<>$ stands for the inner product of the vectors. The difference between the left-hand side and the right-hand side of Eq. (9) is expected to approach zero, typically with at least 13 significant digits. The test results show that term $< \mathbf{H}(\delta x), \mathbf{H}(\delta x) >$ is equal to 0.100159014620902D-17 (D: double precision), and term $< \mathbf{H}^{\mathrm{T}}\big(\mathbf{H}(\delta x)\big), \delta x >$ is equal to 0.100159014620902D-17. The difference between the two terms is 0.577778983316171D-33, which is achieved with 16 digits of accuracy. As a result, the adjoint check has successfully passed under double precision.

For a tangent linear operator, it is also necessary to verify the correctness of the gradient (gradient check) using the following standard:

$$\Phi(\alpha) = \frac{J(\mathbf{c_v}+\alpha)-J(\mathbf{c_v})}{\alpha\nabla J(\mathbf{c_v})}, \tag{10}$$

$$\lim_{\alpha\to0} \Phi(\alpha)=1, \tag{11}$$

where $\nabla J$ is the gradient of $J$ and the symbol $\alpha$ indicates a small scalar value. For values of $\alpha$ that are near but not too close to the machine zero, the value of $\Phi(\alpha)$ is expected to be close to 1. The results of the gradient check are presented in Table 1, showing a satisfactory approximation of the gradient with 8 digits of accuracy achieved ($\alpha=10^{-7}$). This suggests that the tangent linear operator is accurate within the rounding error of the computer.

Table 1. Verification of gradient correctness: values of $\Phi(\alpha)$ for different $\alpha$ values (symbols defined in Eq. (10)).

| $\alpha$ | $\Phi(\alpha)$ |
|---|---|
| $10^{-4}$ | 1.00000684582308 |
| $10^{-5}$ | 1.00000068454433 |
| $10^{-6}$ | 1.00000006939151 |
| $10^{-7}$ | 1.00000000569911 |
| $10^{-8}$ | 1.00000003421803 |
| $10^{-9}$ | 1.00000055706492 |
| $10^{-10}$ | 1.00000626084813 |
| $10^{-11}$ | 1.00001576715311 |
| $10^{-12}$ | 1.00053861393468 |
| $10^{-13}$ | 0.998162037654562 |

8. On page 9, section 4.1.2 Results, it can be seen that when the precipitation threshold is 20mm h-1, the enhancement effect of the experimental group on the forecast is no longer obvious compared to the precipitation threshold of 1mm h-1 and 5mm h-1, and the experiments in the section 4.2 The batch experiment also show a similar situation. Therefore, this phenomenon can be elaborated.

**Response:** Thank you for your suggestion. Based on the previous feedback from other reviewers, we have rerun the batch experiments (considering more assimilation cycles). Additionally, to ensure consistency, the individual case is derived from the batch experiments. The phenomenon is no longer apparent from Figures 4-6 in the revised manuscript v1.

9. The case of heavy precipitation analysis selected in the section 4 Validation could be explained a bit more. That is, why was this case chosen, what is its particularity, and what are its impacts? In particular, what are the characteristics of the vertical velocity during the evolution of this case?

**Response:** Thank you very much for your comments and professional advice. Your concern about the individual case has been raised by another reviewer previously, suggesting that we use an individual case from the batch experiments for consistency. Therefore, we have replaced the individual test with a case in the batch experiments, specifically the case initialized on July 9, 2020 in our revised manuscript v1.

10. The structure of the paper needs to be organized more logically, e.g. on page 6, the text between the section 4 Validation and the section 4.1 The heavy rainfall case study would be better suited to a separate section. Alternatively, there needs to be a separate section on the experimental setup, where the details of section 4.1 and 4.2 related to the setup are described together, which would be more conducive to enhancing the readability of the paper.

**Response:** Thank you for your suggestion. The text between the Section 4 Validation

and the Section 4.1 (which is now the text from Lines 177–194 in the revised manuscript) has been separated into Section 4.1 as suggested. Since in the revised manuscript, the individual case is derived from the batch experiments, we have also adjusted the order of the batch experiments and individual case study, with batch experiments numbered as Section 4.2 and case study as Section 4.3, thus enhancing the readability of the article.

We would like to express our sincere gratitude to you for your invaluable feedback and insightful comments, which have greatly contributed to improving this manuscript.

---

## Author Response (AR3)

**Response to reviewers of the manuscript**

*"A 3D-Var Assimilation Scheme for Vertical Velocity with the CMA-MESO v5.0"*

H. Li, Y. Yang, J. Sun, Y. Jiang, R. Gan, and Q. Xie

for Geoscientific Model Development

**Response to Reviewer #1**

Reviewer #1

The authors have addressed all my comments. I suggest to publish it after a minor revision.

**Response:** Thanks very much for taking your time to review our manuscript. I really appreciate all your comments and suggestions. Please find my responses in the following and my revisions/corrections in the resubmitted files.

Line 20-23: The FSS and ETS statements can be merged.

**Response:** Thank you for your valuable comments. The statements have been revised to "Further assimilation of $w$, in addition to the assimilation of conventional and radial wind data, significantly improves the forecast accuracy of precipitation, resulting in higher FSS (frequency skill score) values and higher ETS (equitable threat score) skills at higher thresholds (5 and 20 mm h$^{-1}$)." in the revised manuscript (Lines 20–22).

Line 318-320: It would be better to say "The adjustments in temperature and humidity increments are achieved by weak physical constraints" first and then pose a potential solution to introduce multivariate correlation in the static BEC. Also, need some references here regarding the multivariate correlation.

**Response:** Thank you for your suggestion. It has been revised to "1) The adjustments in temperature and humidity increments are achieved by weak physical constraints, and it would be better to take into account the multivariate correlation between control variables (Hollingsworth and Lönnberg, 1986; Barker et al., 2004)." as suggested (Lines 319–321 in the revised manuscript).

Line 323: it would be better if say "assimilating w from these observations".

**Response:** Thank you for your suggestion. It has been revised as suggested (Lines 323–324 in the revised manuscript).

We thank the reviewer again for taking the time to review our manuscript.

**Response to Reviewer #2**

Reviewer #2

The authors have addressed most comments, and the quality of the manuscript improves a lot. I suggest publishing it after a minor revision.I am grateful for the chance to review the updated manuscript. The author has adequately addressed my previous concerns and suggestions, leading to a more organized and articulate structure, as well as improved language expression. I think the manuscript is now ready for publication after some additional language refinement as follows.

**Response**: We sincerely thank the reviewer for thoroughly reviewing our manuscript and providing very helpful comments to guide our revision.

1. Line 20: "The further assimilation....", delete "The"; "in addition to the conventional and radial wind data assimilation" revised to "in addition to the assimilation of......".

**Response**: Thank you for your suggestion. It has been revised as suggested (Line 20 in the revised manuscript).

2. Line 29: Change "defining" to "the definition of".

**Response**: Thank you for your suggestion. It has been revised as suggested (Line 28 in the revised manuscript).

3. Line 35: Revise "in enhancing the forecast accuracy of convective precipitation" to "for improving the accuracy of convective precipitation forecast".

**Response**: Thank you for your suggestion. It has been revised as suggested (Line 34 in the revised manuscript).

4. Line 70: Change "to other model state variables for minimizing" to "to other state variables of the model to minimize".

**Response**: Thank you for your suggestion. It has been revised as suggested (Line 69 in the revised manuscript).

5. Line 83: Change "nonhydrostatic" to "non-hydrostatic".

**Response**: Thank you for your suggestion. Revised as suggested (Lines 81–82 in the revised manuscript).

6. Line 113: Change "under the terrain-following vertical coordinate" to "under the vertical coordinate that follows the terrain".

**Response**: Thank you for your suggestion. It has been revised as suggested (Line 113 in the revised manuscript).

7. Line 115: "links the w variable to the u, v...", change "to" to "with".

**Response**: Thank you for your suggestion. Revised as suggested (Line 115 in the revised manuscript).

8. Line 128 : Add "the" before "term".

**Response**: Thank you for your suggestion. Revised as suggested (Line 128 in the revised manuscript).

9. Line 146: Change "The background field's w value" to "The w value of the background field".

**Response**: Thank you for your suggestion. It has been revised as suggested (Line 146 in the revised manuscript).

10. Line 160: Change "It is worth noting" to "It should be noted".

**Response**: Thank you for your suggestion. Revised as suggested (Line 160 in the revised manuscript).

11. Line 161: It should be better to say "there are currently no constraints on the propagation of the impact of the w assimilation in the vertical direction".

**Response**: Thank you for your suggestion. It has been revised as suggested (Lines 160–161 in the revised manuscript).

12. Line 162: Change "thus leading" to "which leads".

**Response**: Thank you for your suggestion. Revised as suggested (Line 162 in the revised manuscript).

13. Lines 178-179: Change "the assimilation experiment's scope" to "the scope of the assimilation experiment".

**Response**: Thank you for your suggestion. It has been revised as suggested (Lines 178–179 in the revised manuscript).

14. Line 179: Change "that" to "what".

**Response**: Thank you for your suggestion. Revised as suggested (Line 179 in the revised manuscript).

15. Line 190: Change "the maximum w value" to "the maximum value of w".

**Response**: Thank you for your suggestion. Revised as suggested (Line 190 in the revised manuscript).

16. Line 207: Change "run" to "ran".

**Response**: Thank you for your suggestion. Revised as suggested (Line 207 in the revised manuscript).

17. Line 236: "demonstrate" should be "demonstrates".

**Response**: Thank you for your suggestion. Revised as suggested (Line 236 in the revised manuscript).

18. Line 268: Change "The heavy precipitation center" to "The center of heavy precipitation".

**Response**: Thank you for your suggestion. Revised as suggested (Line 268 in the revised manuscript).

19. Line 300: Change "assimilating" to "the assimilation of".

**Response**: Thank you for your suggestion. Revised as suggested (Line 300 in the revised manuscript).

20. Line 327: Change "given that" to "since".

Response: Thank you for your suggestion. Revised as suggested (Line 328 in the revised manuscript).

I commend the authors for their efforts and cooperation, and look forward to seeing the article published.

We appreciate the warm work of the reviewer in earnest and hope that the correction will meet with approval. Once again, thank you very much for your comments and suggestions.